**METHOD**

# HyperChIP: identification of hypervariable signals across ChIP-seq or ATAC-seq samples

Haojie Chen[1,2†], Shiqi Tu[1*†], Chongze Yuan[3], Feng Tian[1,2], Yijing Zhang[4], Yihua Sun[3] and Zhen Shao[1,2*]

*Correspondence:
tushiqi@picb.ac.cn;
shaozhen@picb.ac.cn
†Haojie Chen and Shiqi Tu
contributed equally to this
work.
[1] CAS Key Laboratory
of Computational
Biology, Shanghai
Institute of Nutrition
and Health, Chinese
Academy of Sciences,
Shanghai 200031, China
[2] University of Chinese
Academy of Sciences,
Beijing 100049, China
Full list of author information
is available at the end of the
article

## Abstract

Identifying genomic regions with hypervariable ChIP-seq or ATAC-seq signals across given samples is essential for large-scale epigenetic studies. In particular, the hypervariable regions across tumors from different patients indicate their heterogeneity and can contribute to revealing potential cancer subtypes and the associated epigenetic markers. We present HyperChIP as the first complete statistical tool for the task. HyperChIP uses scaled variances that account for the mean-variance dependence to rank genomic regions, and it increases the statistical power by diminishing the influence of true hypervariable regions on model fitting. A pan-cancer case study illustrates the practical utility of HyperChIP.

**Keywords:** ChIP-seq, ATAC-seq, Hypervariable regions, Epigenetic heterogeneity, Large-scale cancer studies

## Background

Chromatin immunoprecipitation followed by high-throughput sequencing (ChIP-seq) is the premier technology for profiling genome-wide localization of chromatin-binding proteins, including transcription factors and histones with various modifications [1, 2]. Besides, assay for transposase-accessible chromatin using sequencing (ATAC-seq) has been widely adopted for the detection of open chromatin [3]. As a common computational task for learning from ChIP/ATAC-seq data, identifying genomic regions with significant changes of ChIP/ATAC-seq signal intensities across samples is essential to understanding the epigenetic mechanisms that orchestrate the variation of gene expression program [4–6]. For this task, there are two major analyses suited to distinct application scenarios. The first one is referred to as differential analysis, in which the label of each sample is clearly defined (e.g., healthy or diseased) and differential ChIP/ATAC-seq signals between different labels are identified by comparing the corresponding samples [7, 8]. The second analysis does not require prior knowledge regarding the labels of samples and aims at identifying hypervariable ChIP/ATAC-seq signals across samples, which can then be used as features for the classification of the samples. This analysis is

of particular importance in classifying samples of different cancer patients and gaining insights into the epigenetic markers of different cancer subtypes/stages [6, 9, 10]. Since the second analysis is intrinsically an unsupervised one, it typically requires many more samples to achieve reliable results compared to the first analysis.

In the early years, the practical applicability of hypervariable analysis was seriously limited by the number of ChIP-seq samples available in a study, and researchers were more inclined to apply differential analysis with a proper experimental design. As a result, a large body of mature computational tools has been developed for differential ChIP-seq analysis [7, 8]. A contrasting example is the analysis of single-cell RNA-seq (scRNA-seq) data. As an individual scRNA-seq experiment generates transcriptome profiles of a large number of cells and researchers typically have no detailed prior knowledge about the cell identities, hypervariable analysis has been frequently applied to scRNA-seq data [11–13]. Accordingly, many computational tools for identifying hypervariable genes (HVGs) from scRNA-seq data and using these genes to classify cells have been developed [14–16].

In recent years, with the decrease of sequencing costs, there are more and more large-scale studies in which tens or even hundreds of ChIP/ATAC-seq profiles for different human individuals are generated, and hypervariable analysis is becoming increasingly prevalent in the analysis of ChIP/ATAC-seq data [6, 9, 10, 17, 18]. In particular, hypervariable ChIP/ATAC-seq signals across cancer patients could be potential epigenetic markers of different cancer subtypes/stages, and these markers may serve as therapeutic targets and may contribute to the prognosis of patients [6]. To our best knowledge, however, there are currently no such computational tools that are specifically developed for hypervariable ChIP/ATAC-seq analysis.

In many studies, researchers designed their own computational pipelines for calling hypervariable ChIP/ATAC-seq signals [9, 10, 17, 19–22], but some of these pipelines failed to take some basic data characteristics into account. For example, sequencing count data are inherently associated with a strong dependence of signal variability on the mean signal intensity, making the ChIP/ATAC-seq signal variability of different genomic regions not directly comparable with each other. Specifically, after a logarithmic transformation, small log-counts tend to have larger variances than large log-counts [23, 24]. While this mean-variability relationship has been properly accounted for in almost all the tools for calling HVGs from scRNA-seq data, several studies called hypervariable ChIP/ATAC-seq signals by ranking genomic regions based on some variability index without considering its dependence on the mean signal intensity [9, 10, 19, 21, 22]. Other studies alleviated the influence of the mean-variability trend by applying various practical strategies, such as combining the rankings based on the mean intensities and variances [20] and making a log-transformation with a large offset count to suppress the large variances of small log-counts [17]. These strategies were effective, but their implementation details (e.g., the exact offset count) and performance highly rely on the specific data set, and thus, their general applicability is questionable.

A more fundamental problem is that, instead of coming up with a probabilistic model to assess the statistical significances of observed ChIP/ATAC-seq signal variability, all these studies ranked genomic regions and selected a certain number or proportion of top-ranked ones as hypervariable regions (HVRs), with the specific

number or proportion being determined based on practical experience. In these studies, the numbers and proportions ranged from 1000 to 10,000 and from 1 to 25%, respectively. On the one hand, designing a complete statistical model with *p*-value calculation can increase the adaptivity to various data sets and avoid an arbitrary selection of HVRs. On the other hand, using an ordinary model fitting framework will almost certainly lead to conserved *p*-values and low statistical power for identifying HVRs, since it is difficult to strictly avoid the influence of true HVRs on the fitting process. A previous study of ours has suggested that *p*-values derived from an ordinary parameter estimation framework can hardly bear the strength of multiple testing adjustment [6].

In this study, we present HyperChIP, a statistical method for hypervariable ChIP/ATAC-seq analysis that is aimed at addressing the above concerns. In this method, a variability statistic that has been corrected for the mean signal intensity is used as the key statistic, and specific efforts have been made to increase the statistical power for identifying HVRs. For the latter, HyperChIP selects a subset of genomic regions with relatively low signal intensities for parameter estimation and further employs the winsorization technique [25] to render the estimation procedure robust to true HVRs. Our empirical observations on various data sets suggest that these low-intensity regions contain only a small proportion of HVRs, which can be effectively handled by winsorization. Applying HyperChIP to several real data sets, of which each comprised ChIP/ATAC-seq profiles of tens of cancer patients, we found that the method can identify hundreds to thousands of significant HVRs at common cutoffs of the BH-adjusted *p*-value, which controls the false discovery rate [26]. Further exploration revealed that the identified HVRs tended to be associated with the tumor progression stages of patients. We also observed a systematic difference in variability structure between proximal and distal regions. Specifically, the ChIP/ATAC-seq signal variability in distal regions was considerably higher than that in proximal regions, which was consistent with previous studies showing that the activity of enhancer elements is much more variable across individuals and cellular contexts than is the activity of gene promoters [27–29]. We therefore highlight the necessity of separately dealing with proximal and distal regions in a hypervariable ChIP/ATAC-seq analysis, to avoid the suppression of the statistical power for identifying proximal HVRs.

To demonstrate the practical utility of HyperChIP, we additionally applied it to a pan-cancer ATAC-seq data set, which was generated by The Cancer Genome Atlas (TCGA) program and consisted of ATAC-seq profiles of tumors from hundreds of patients across 23 cancer types [17]. Based on the identified HVRs, we investigated the similarity structure among the ATAC-seq profiles. While most of them were well clustered by their cancer types, those of various types of squamous cell carcinoma (SC) tended to be mixed up with each other. We further defined super classes of cancer types based on the similarity structure and identified transcription factors (TFs) specific to each class by applying a motif-scanning procedure on the HVRs. Notably, many of the identified TFs were found to be exclusively expressed in the corresponding classes showing a strong association with either tissue specificity, tumorigenesis, or both. For example, *TP63*, a confirmed oncogene in several SC types [30, 31], was identified as the most significant TF specific to the SC class.

## Results

### Ranking genomic regions based on scaled variances

To evaluate the performance of HyperChIP, we collected three data sets from large-scale cancer studies (Table 1). The first data set comprised H3K27ac ChIP-seq profiles, a histone modification marking both active promoters and enhancers, of tumor tissues of 36 lung adenocarcinoma (LUAD) patients [6]. The second one consisted of ATAC-seq samples of 34 non-small cell lung carcinoma (NSCLC) patients [18]. The third one consisted of RNA polymerase (Pol) II ChIP-seq samples of 26 LUAD cell lines derived from different patients [32].

To facilitate the understanding of how HyperChIP works, we briefly describe the required input data of it. HyperChIP takes a matrix of normalized $\log_2$ read counts as input. The rows and columns of the matrix correspond to a pre-defined list of genomic regions and a set of ChIP-seq samples, respectively. In this study, we separately compiled a list of regions for each data set. Given a data set, we first called peaks for each sample and merged all the resulting peaks. Broad merged peaks were then divided up into consecutive bins, and narrow ones were left as they were. As for normalization, we constructed a pseudo-reference profile by averaging all the samples and invoked the MA normalization procedure implemented in MAnorm2 [24] to normalize each sample against it (see the "Methods" section). Note also that, unless otherwise stated, each hypervariable analysis in this study targeting an individual data set has separately handled proximal and distal regions.

Given a matrix of normalized signal intensities, HyperChIP accounts for the associated mean-variability relationship by applying a gamma family regression method to observed mean-variance pairs. Specifically, it employs a local regression procedure to allow for general mean-variance relationships [33]. It can be easily seen that all the three data sets are associated with clear mean-variance dependence, and the specific trend depicted by the fitted mean-variance curve (MVC) varies across the data sets (Fig. 1a; Additional file 1: Fig. S1). Then, HyperChIP uses the MVC to derive a scaled variance for each genomic region, which is defined as the ratio of the observed variance to the predicted variance obtained from the MVC (Fig. 1b; Additional file 1: Fig. S2). These scaled variances shall be used for ranking regions and selecting HVRs.

To benchmark HyperChIP, we considered several other methods for ranking regions. These methods can be classified into two classes. One class uses some variability statistics to rank regions without taking its dependence on the mean signal intensity into account. Such statistics included the observed (unscaled) variance [19, 21, 22], median absolute deviation (MAD) [9], and interquartile range (IQR) [10]. We found that all the methods of this class were associated with a tendency to select regions with low signal

**Table 1** Large-scale cancer data sets used for benchmarking HyperChIP. Each data set was comprised of ChIP/ATAC-seq profiles of tens of cancer patients

| Data set | Biological context | Cohort size | Source |
| --- | --- | --- | --- |
| H3K27ac ChIP-seq | Tumor tissues of LUAD patients | 36 | Yuan et al. [6] |
| ATAC-seq | Tumor tissues of NSCLC patients | 34 | Wang et al. [18] |
| Pol II ChIP-seq | Cancer cell lines derived from LUAD patients | 26 | Suzuki et al. [32] |

*LUAD* Lung adenocarcinoma, *NSCLC* Non-small cell lung carcinoma

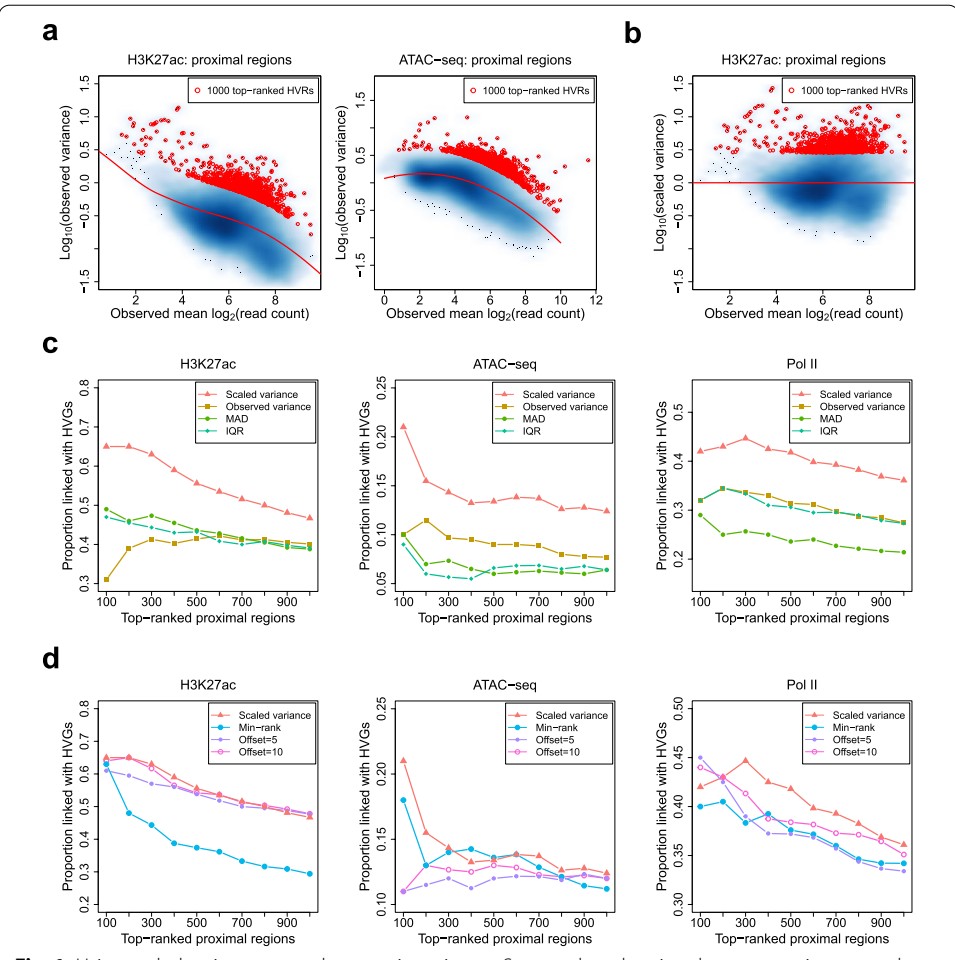

**Fig. 1** Using scaled variances to rank genomic regions. **a** Scatter plots showing the mean-variance trends associated with different data sets. Variance is shown at the $\log_{10}$ scale. Red lines depict the corresponding MVCs. Red points mark the 1000 regions with the largest scaled variances. **b** A scatter plot of $\log_{10}$ scaled variances against observed mean signal intensities. **c**, **d** For each data set in Table 1, the TDP among top-ranked proximal regions is plotted against the number of top-ranked proximal regions for each method. MAD, median absolute deviation; IQR, interquartile range

intensities as HVRs (Additional file 1: Fig. S3a). The methods of the other class adopt different practical strategies to account for the mean-variability dependence. We included two methods in this class. The first method, referred to as min-rank, is primarily aimed at selecting regions with both high intensities and high variability [20]. It first separately uses the observed mean intensities and the observed variances to sort genomic regions into ascending order. Then, it takes the minimum of the two ranks associated with each region and uses these minima to once again rank the regions. When applying min-rank, we found this method clearly tended to select regions with high intensities as HVRs (Additional file 1: Fig. S3b). The second method follows a computational pipeline used in a previous study [17]. It first derives count per million (CPM) values and applies a $\log_2$ transformation with a moderately large offset. The results are then subject to quantile normalization. For this method, we have separately tried 5 and 10 as the offset value (Additional file 1: Fig. S3c).

To assess the rankings of genomic regions provided by each method, we calculated true discovery proportions (TDPs) among top-ranked regions. More specifically, we defined, among proximal regions, true HVRs as those regions that were linked with HVGs, which were identified based on the corresponding RNA-seq data (see the "Methods" section and Additional file 1: Fig. S4). We then plotted the TDP against the number of the top-ranked proximal regions for each method. Compared with the methods that do not consider the mean-variability dependence, HyperChIP achieved much higher TDPs for all the three data sets (Fig. 1c). Compared with the methods of the other class, HyperChIP performed better or as well as them depending on the specific data set (Fig. 1d).

A common downstream analysis based on identified HVRs is to use them as features for the classification of samples, with the hope of revealing the substructure of the samples. In the ATAC-seq data set, the 34 NSCLC patients consisted of 26 LUAD and 8 LUSC (lung squamous cell carcinoma) cases, corresponding to two primary subtypes of NSCLC [18]. These subtype labels can be considered the gold standard for evaluating the classifications of the patients. Here, based on the top-ranked HVRs derived by different methods, we performed classifications of the patients into two subgroups (see the "Methods" section) and assessed the consistency between the classifications and the gold standard by using the adjusted Rand index (ARI) [34]. This index has an expected value of 0 for random classifications and is bounded above by 1 for a perfect agreement between two classifications. The relative performance of different methods was similar as in the previous comparison based on HVGs: HyperChIP showed an overall performance better than each of the other methods, especially when it was compared with the methods that do not consider the mean-variability dependence (Fig. 2a, b). Furthermore, the classification results from HyperChIP were very stable across a wide range of numbers of used HVRs (Fig. 2c).

### Modeling scaled variances and increasing the statistical power for identifying HVRs

A complete statistical model specifying the null distribution of the scaled variances is required for assessing their statistical significances. We previously developed MAnorm2 for differential ChIP-seq analysis [24]. In HyperChIP, we follow the distributional theory proposed by MAnorm2 for modeling a group of ChIP-seq samples. Specifically, we assume the normalized $\log_2$ read counts at each genomic region follow a normal distribution, with the precision parameter associated with a prior gamma distribution whose expectation value is determined by the MVC. Formally, let $X_{ij}$ denote the normalized $\log_2$ read count at region $i$ in sample $j$. We assume:

$$X_{ij} \mid \sigma_i^2 \sim N\left(\mu_i, \gamma \sigma_i^2\right), \tag{1}$$

$$\frac{1}{\sigma_i^2} \sim \frac{1}{f(\mu_i)} \cdot \frac{\chi_{d_0}^2}{d_0}. \tag{2}$$

Here, $\mu_i$ and $\sigma_i^2$ are two unknown parameters that quantify the mean signal intensity at region $i$ and the associated signal variability, respectively; $f(\cdot)$ denotes the MVC; $d_0$, referred to as the number of prior degrees of freedom, effectively assesses how dispersedly the observed variances are distributed around the MVC (larger values of $d_0$ indicate the

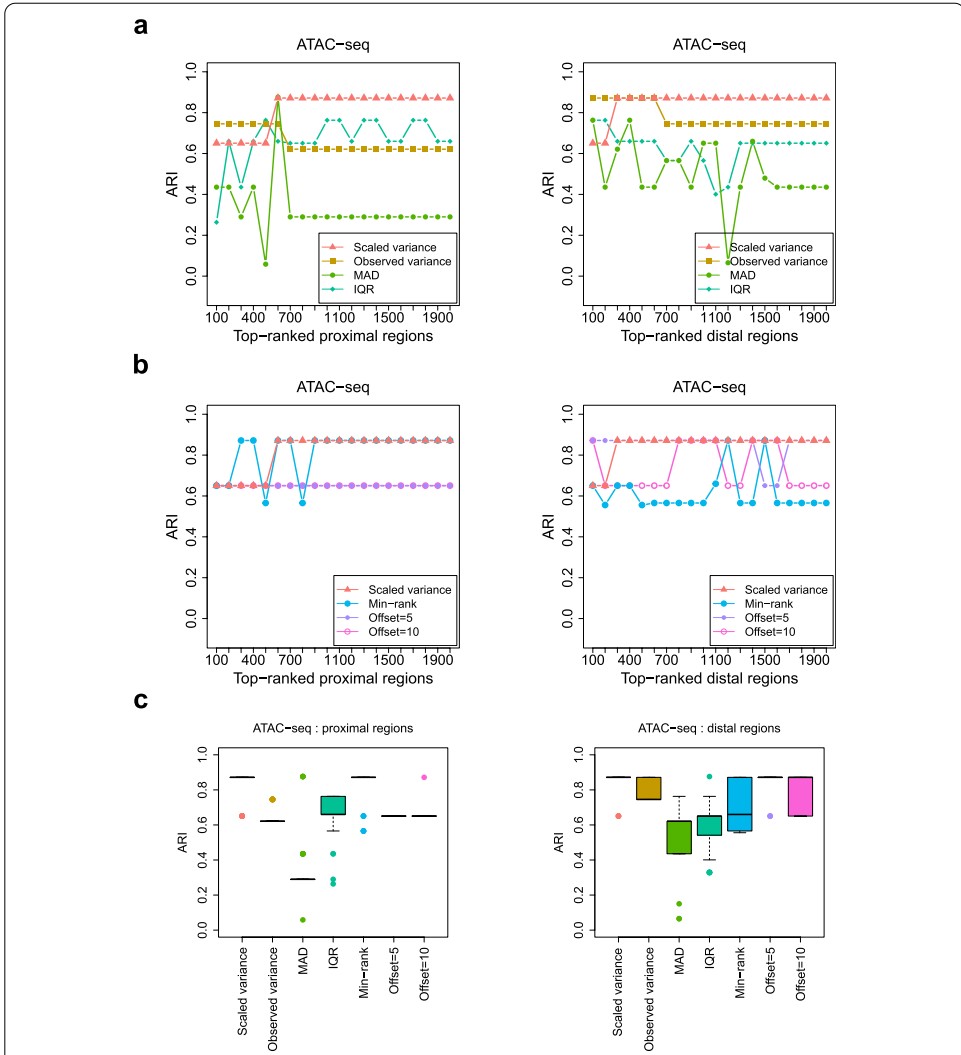

**Fig. 2** Evaluating the classifications of samples that are based on top-ranked HVRs identified by different methods. **a**, **b** For each method applied to the ATAC-seq data set, the ARI is plotted against the number of top-ranked HVRs that are used to classify the ATAC-seq samples. Each ARI value assesses the consistency between the classification and the true labels of the samples. Proximal and distal regions are separately analyzed and ranked. **c** Box plots for the ARI values resulting from using a wide range of numbers of top-ranked HVRs to classify the samples. For each method, the sequence of numbers starts with 50 and ends with 5000, with an increment of 50

observed variances are more concentrated at the MVC); $\gamma$ is referred to as the variance ratio factor and is designed for scaling the MVC to better fit the observed mean-variance pairs under this model formulation.

Next, suppose $\hat{\mu}_i$ and $\hat{t}_i$ are the observed mean intensity and variance of region $i$, respectively (i.e., the sample mean and sample variance of all $X_{ij}$ associated with region $i$). Based on the model, it follows:

$$\frac{\hat{t}_i}{f(\mu_i)} \sim \gamma F_{m-1,d_0},$$

(3)

where $m$ is the total number of samples. The scaled variances for ranking genomic regions are exactly obtained by replacing $\mu_i$ on the left-hand side of (3) with $\hat{\mu}_i$, and HyperChIP uses the right-hand side of (3) as the null distribution of the scaled variances for deriving $p$-values (i.e., upper-tailed probabilities). Additional file 2: Note S1 provides a justification for this approximation by performing a random simulation.

Previously, we adopted in MAnorm2 a method of moments for parameter estimation, in which $d_0$ and $\gamma$ are estimated by matching the first two sample moments of log-scaled variances with the corresponding theoretical moments of $\log\left(\gamma F_{m-1,d_0}\right)$. In a hypervariable analysis, however, using this method would lead to low statistical power for identifying HVRs, since true HVRs are included in the calculation of sample moments as well and their large variances would give rise to an underestimated $d_0$. In fact, when applying this method to the three data sets, we can barely identify any significant HVRs at common cutoffs of the BH-adjusted $p$-value (Tables 2 and 3).

This low-power problem can be alleviated by selecting a subset of genomic regions with a relatively low abundance of HVRs to estimate the parameters, provided that the selection criterion is statistically independent of the test statistic (i.e., the scaled variance) under the null model. Inspired by the independent filtering strategy designed by DESeq2 for reducing the strength of multiple testing adjustment [35], HyperChIP uses the observed mean intensity of each region as its selection criterion. To determine the specific criterion, we first examined how top-ranked HVRs were distributed along the

**Table 2** Numbers of significant proximal HVRs identified by applying different parameter estimation methods with various cutoffs of the BH-adjusted $p$-value. The original method refers to the moment matching method used by MAnorm2, and the other method corresponds to the default settings of HyperChIP

| Parameter estimation | Data set | Number of significant proximal HVRs | | |
|---|---|---|---|---|
| | | 0.01 | 0.05 | 0.1 |
| Original | H3K27ac ChIP-seq | 0 | 2 | 2 |
| | ATAC-seq | 5 | 8 | 10 |
| | Pol II ChIP-seq | 0 | 0 | 0 |
| Lower 10% + winsorization | H3K27ac ChIP-seq | 237 | 789 | 1442 |
| | ATAC-seq | 1050 | 2171 | 3197 |
| | Pol II ChIP-seq | 303 | 990 | 1546 |

**Table 3** Numbers of significant distal HVRs identified by applying different parameter estimation methods with various cutoffs of the BH-adjusted $p$-value

| Parameter estimation | Data set | Number of significant distal HVRs | | |
|---|---|---|---|---|
| | | 0.01 | 0.05 | 0.1 |
| Original | H3K27ac ChIP-seq | 2 | 8 | 12 |
| | ATAC-seq | 0 | 3 | 4 |
| | Pol II ChIP-seq | 0 | 0 | 0 |
| Lower 10% + winsorization | H3K27ac ChIP-seq | 289 | 1372 | 2621 |
| | ATAC-seq | 3124 | 6830 | 10,234 |
| | Pol II ChIP-seq | 602 | 1400 | 2122 |

range of mean intensities for each of the three data sets. More specifically, we sorted all regions in ascending order (with respect to the mean intensities) and divided them into 10 equally sized groups. We then calculated the proportion of those regions among each group that were ranked in the top 2000 HVRs. It was found that, among proximal regions, top-ranked HVRs tended to be enriched within the groups with moderately large mean intensities, while for distal regions, they tended to be enriched within the groups with the largest mean intensities (Fig. 3a; Additional file 1: Fig. S5a). We also noticed that previous studies had observed many genes with very stable expression strength across cellular contexts. These genes, referred to as lowly variable genes (LVGs), tended to be related to fundamental or so-called constitutive cellular processes, such as translation and translational elongation [15]. Since the regions in which ChIP-seq signals have extremely low variability across samples could also lead to the underestimation of $d_0$, we examined as well the distribution of top-ranked lowly variable regions (LVRs), namely the regions having the smallest scaled variances (Fig. 3b). For both proximal and distal regions, we found that the top-ranked LVRs tended to be enriched within the groups with the largest mean intensities (Fig. 3c; Additional file 1: Fig. S5b). For all the three data sets, the only exception was that the top-ranked proximal LVRs associated with the Pol II ChIP-seq data set were enriched within the center-right groups as well as within the rightmost group, showing a bimodal distribution profile (Additional file 1: Fig. S6).

Based on these observations, we set the default behavior of HyperChIP to using the 10% of regions having the smallest mean intensities for parameter estimation. To further account for the presumably small amount of HVRs and LVRs among these low-intensity regions, HyperChIP integrated the winsorization procedure [25] into the original moment matching method to avoid their influence on parameter estimation (see the "Methods" section). Using the three data sets as examples, we tried different parameter estimation strategies and compared the resulting $d_0$ estimates. With the subset selection and the use of winsorization, a stepwise increase in the estimated $d_0$ was consistently observed across the data sets (Fig. 3d; Additional file 1: Fig. S5c), suggesting an improved statistical power associated with the model fitted by HyperChIP.

To more directly demonstrate the effect of the modifications made by HyperChIP to the original MAnorm2 method, which does not apply a subset selection and winsorization, we compared the empirical distribution of scaled variances with the null distributions inferred by the two methods. We first paid specific attention to the low-intensity regions selected by HyperChIP for parameter estimation, and we observed that, when applying the original method, most of these regions did not even reach the variability magnitude as

(See figure on next page.)

**Fig. 3** Selecting a subset of genomic regions and using winsorization for parameter estimation. **a** For the H3K27ac ChIP-seq data set, bar plots show the distributions of top-ranked proximal/distal HVRs along the range of mean intensities. Proximal and distal regions have been separately divided into 10 equally sized groups based on the observed mean signal intensities. **b** Scatter plots of $\log_{10}$-scaled variances against the mean intensities of proximal/distal regions. Top-ranked HVRs and LVRs are separately highlighted. **c** Bar plots showing the distributions of top-ranked proximal/distal LVRs along the range of mean intensities. **d** $d_0$ estimates resulting from different parameter estimation methods. The original method refers to the moment matching method employed by MAnorm2, which does not apply a subset selection and winsorization. The method labeled as "lower 10%" uses only the 10% of regions having the smallest mean intensities for parameter estimation

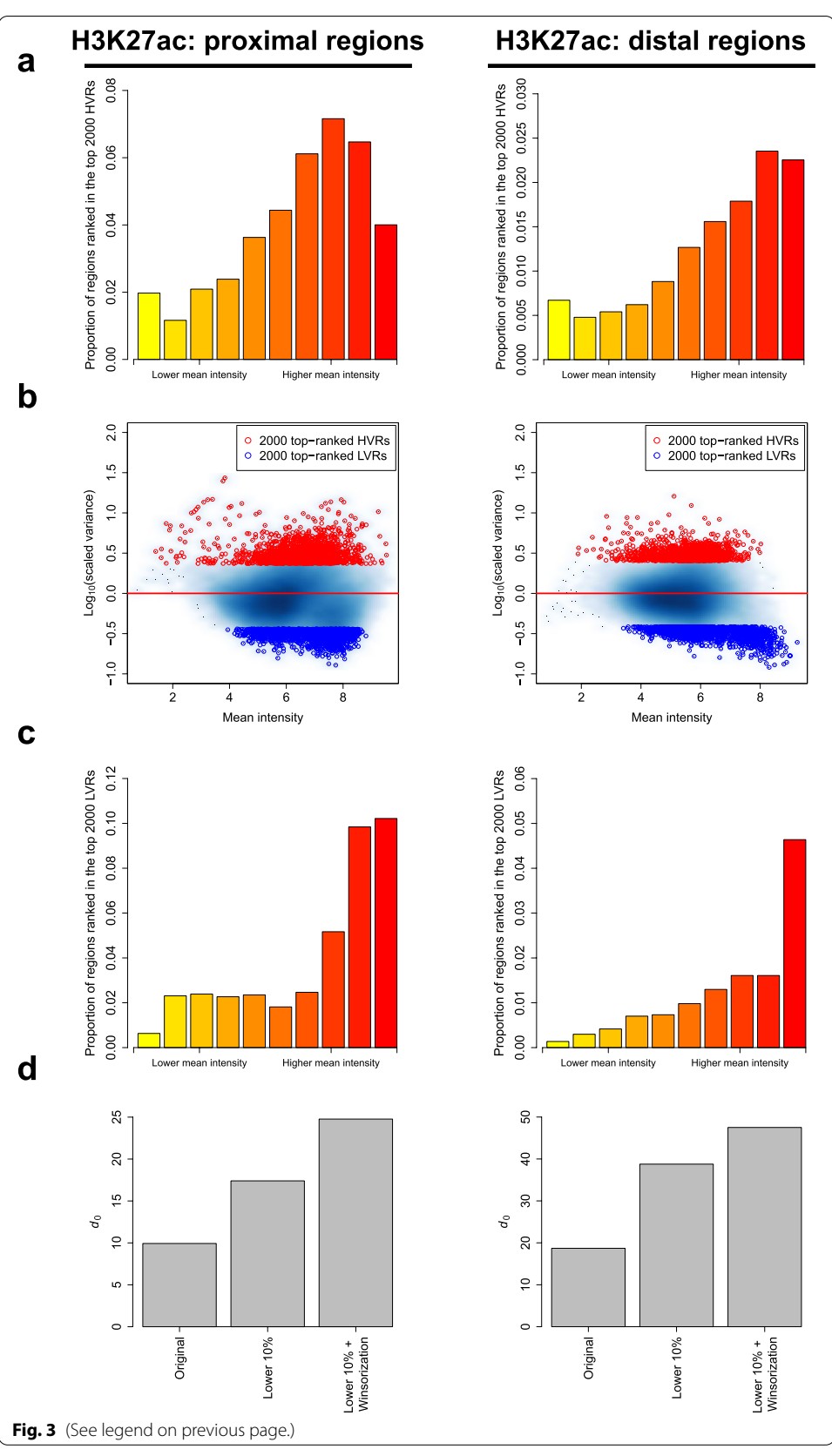

**Fig. 3** (See legend on previous page.)

suggested by the null distribution (Fig. 4a), indicating this null distribution was too biased towards large values for sensitive identification of HVRs. In comparison, the null distribution inferred by HyperChIP better matched these regions, especially for the regions with relatively small-scaled variances (Fig. 4b). Accompanying this improved fit to the low-intensity regions was a dramatic change of the overall distribution of *p*-values (Fig. 4c, d). Specifically, the *p*-values resulting from the original method were generally conserved with no enrichment near 0, while the *p*-values derived by HyperChIP showed clear enrichments at both ends of [0, 1], highlighting improved statistical power for identifying LVRs as well as HVRs. With this improvement, we were now able to detect hundreds to thousands of statistically significant HVRs for each of the three data sets (Tables 2 and 3).

### Biological interpretation of HVRs and LVRs

To explore the roles of HVRs/LVRs in specific biological contexts, we first defined a set of significant proximal HVRs/LVRs for the H3K27ac ChIP-seq data set (BH-adjusted

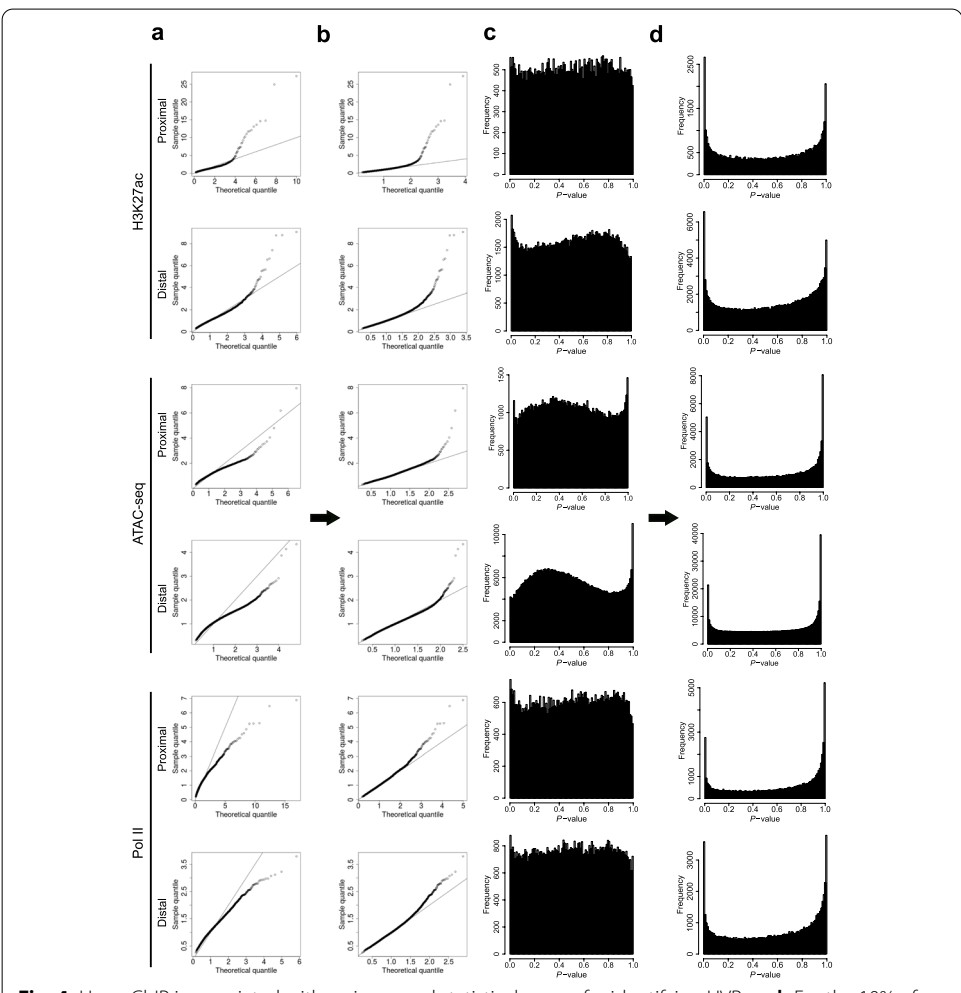

**Fig. 4** HyperChIP is associated with an improved statistical power for identifying HVRs. **a**, **b** For the 10% of genomic regions having the smallest mean intensities, plotting sample quantiles of the scaled variances against theoretical quantiles of the null distribution inferred by **a** the original method or **b** HyperChIP (i.e., the method labeled as "lower 10% + winsorization"). **c**, **d** The overall distributions of (one-tailed) *p*-values resulting from **c** the original method or **d** HyperChIP

*p*-value < 0.1; for the identification of significant LVRs, *p*-values were derived as lower-tailed probabilities of the null distribution). Then, we performed Gene Ontology (GO) enrichment analysis for the genes linked with these HVRs/LVRs (only the GO terms for biological processes were used). For comparison, we also randomly selected a set of other proximal peak regions that matched the number of the HVRs. Compared to the genes linked with these randomly selected regions, from which no significant GO terms were enriched when relatively stringent *p*-value cutoffs were applied, the genes linked with the HVRs/LVRs enriched more terms (Fig. 5a), suggesting they might be associated with coordinated biological functions. Specifically, the genes linked with the LVRs were enriched for GO terms of constitutive cellular processes, including various histone modifications, tRNA modification, and RNA catabolic processes (Fig. 5b; Additional file 3:

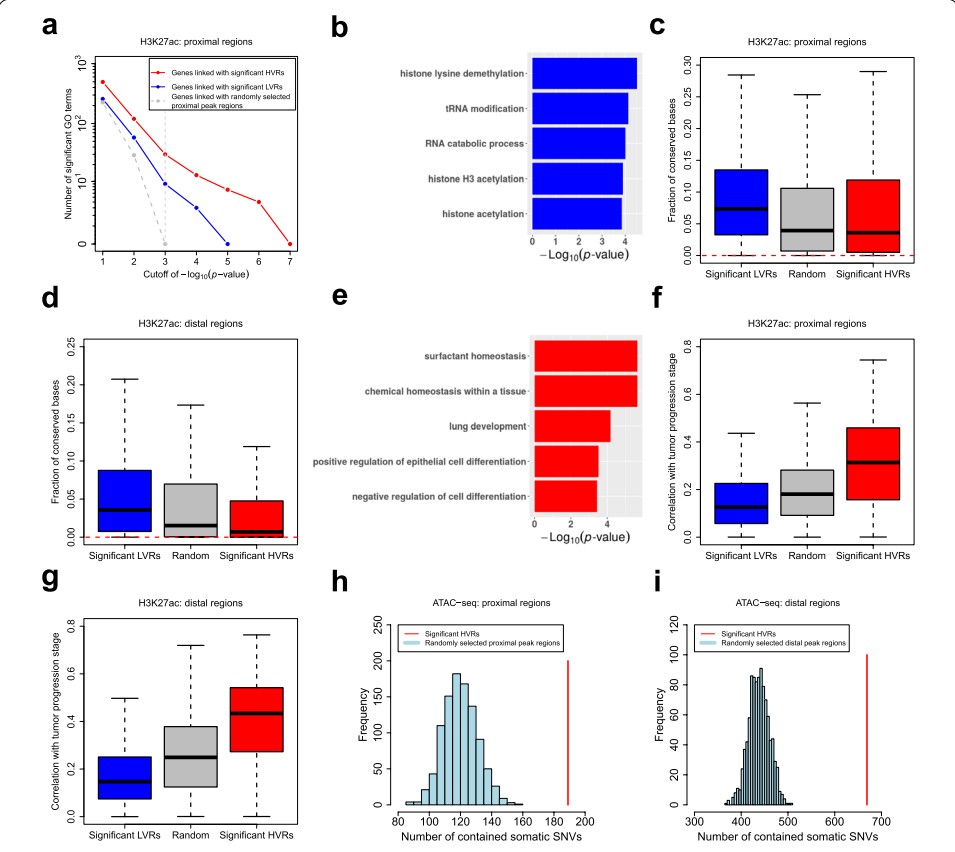

**Fig. 5** Biological interpretation of HVRs and LVRs. **a** For each gene set, the number of significant GO terms is plotted against the *p*-value cutoff used in the GO enrichment analysis. Significant proximal HVRs/LVRs used here are defined based on the LUAD H3K27ac ChIP-seq data set. **b** Example GO terms enriched from the genes linked with the significant proximal LVRs. Additional file 3: Table S1 gives a full list of the top 20 GO terms. **c**, **d** Box plots showing the fractions of conserved bases in different genomic regions. Here, the conserved bases are defined as those with a phastCons score over 0.9. Red dotted lines indicate the genome-wide fraction. **e** Example GO terms enriched from the genes linked with the significant proximal HVRs. Additional file 3: Table S2 gives a full list of the top 20 GO terms. **f**, **g** Box plots showing the correlations of H3K27ac levels in different regions with the clinical stage of LUAD progression. Here, the correlations are measured by the absolute value of the Spearman correlation coefficient. **h** Significant proximal HVRs defined for the NSCLC ATAC-seq data set are enriched with somatic SNVs. We have performed 1000 times of random simulation. In each time, a set of proximal peak regions matching the number of the HVRs has been randomly selected. **i** Significant distal HVRs are enriched with somatic SNVs as well

Table S1). Consistently, the LVRs showed clearly higher sequence conservation across species [36] than both the HVRs and the randomly selected regions (Fig. 5c). Similar results were also observed when we defined significant distal HVRs/LVRs and selected a set of random distal peak regions (Fig. 5d).

In contrast, the genes linked with the proximal HVRs were enriched for GO terms related to lung development, cell differentiation, or the identity of lung cells (Fig. 5e; Additional file 3: Table S2). For example, surfactant homeostasis, which is a kind of chemical homeostasis important for the lungs, is a characteristic biological process of alveolar cells [37], a confirmed cell of origin of LUAD [38, 39]. Since alveolar cells become more and more poorly differentiated and gradually lose their cell identity as LUAD progresses [40], these results implied that the epigenetic heterogeneity at the proximal HVRs across the LUAD patients may be connected with their different tumor progression stages. Following this speculation, we examined the correlations of the H3K27ac ChIP-seq signals in different regions with the histopathological labels of the patients that can reflect their tumor progression stages (see the "Methods" section), and we observed considerably stronger correlations on the proximal HVRs compared to both the proximal LVRs and the random proximal peak regions (Fig. 5f). Similar results were also observed on the distal regions (Fig. 5g). We next evaluated the prognostic associations of different regions by separately performing a Cox regression on the H3K27ac ChIP-seq signal in each region [41]. It was found that the proximal/distal HVRs were more significantly associated with the survival time of the patients than the proximal/distal LVRs and the random proximal/distal peak regions (Additional file 1: Fig. S7a, b). Moreover, we performed a hierarchical clustering of the patients by using both the proximal and distal HVRs as features (see the "Methods" section). The patients were classified into two distinct subgroups, and a significant survival difference was observed between them (Additional file 1: Fig. S7c, d).

Genetic variation across individuals has been revealed as one of the causes of the associated epigenetic heterogeneity [28, 42]. To explore this relationship, we assessed the enrichment of genetic variants within HVRs. Here, we defined significant proximal and distal HVRs for the ATAC-seq data set and mapped them to the somatic single nucleotide variants (SNVs) identified for the associated NSCLC patients by the original study [18]. By random simulation, we observed that both the proximal and distal HVRs contained significantly more somatic SNVs than by chance (Fig. 5h, i). Similarly, the HVRs showed a significant association with somatic copy number variation (Additional file 1: Fig. S8). We also obtained a list of germline single nucleotide polymorphisms (SNPs) for the patients and found they were enriched within the HVRs as well (Additional file 1: Fig. S9). Among these germline SNPs, we next performed a systematic identification of the quantitative trait loci (QTLs) at which different genotypes were associated with significantly differential ATAC-seq signals in the vicinity (see the "Methods" section). Owing to the large number of statistical tests and the relatively small sample size, only several significant QTLs were identified after multiple testing adjustments, and most of them were located within the HVRs (Additional file 1: Fig. S10a, b). For example, the most significant QTL was located within a proximal HVR, and its genotype was significantly associated with the ATAC-seq signal at the HVR as well as the expression strength of the downstream gene (Additional file 1: Fig. S10c, d). The most significant distal QTL

was located within an HVR as well, and its genotype was significantly associated with both the ATAC-seq signal at the HVR and the expression strength of the nearest gene (Additional file 1: Fig. S10e, f). In conclusion, these findings indicated that the epigenetic heterogeneity at the HVRs across the patients was linked with their genetic variation.

### Applying HyperChIP to a large pan-cancer ATAC-seq data set

To illustrate the practical utility of HyperChIP, we applied it to a pan-cancer ATAC-seq data set from TCGA [17], which comprised ATAC-seq samples of tumor tissues of 410 patients across 23 cancer types (Additional file 3: Table S3). Applying HyperChIP to this data set, we identified 5823 proximal HVRs and 2393 distal ones (BH-adjusted $p$-value < 0.1). Based on the ATAC-seq signals in these HVRs, we performed a t-distributed stochastic neighbor embedding (t-SNE) analysis [43] to dissect the similarity structure among the patients (see the "Methods" section). Naturally, most of the cancer types were well separated from each other in the two-dimensional t-SNE plot (Fig. 6a). We indeed, however, noticed a few exceptions. For example, kidney renal clear cell carcinoma (KIRC) and kidney renal papillary cell carcinoma (KIRP), which have the same tissue of origin [44], were very close to each other in the t-SNE plot. Another similar example was glioblastoma multiforme (GBM) and brain lower grade glioma (LGG), both of which were brain cancers. Of note, there were two mixtures of cancer types, both of which involved cancer types of different tissue origins (Fig. 6b, c). Further, we found that the two mixtures largely corresponded to SC and digestive adenocarcinoma (DIAD), respectively. Specifically, the esophageal carcinoma (ESCA) patients in this data set comprised 12 esophageal squamous cell carcinoma (ESSC) and 6 esophageal adenocarcinoma (ESAD) cases, and the distribution of these two subtypes in the t-SNE plot was very consistent with that of the SC and DIAD classes (Fig. 6d), suggesting the HVRs identified by HyperChIP can contribute to revealing the substructures of individual cancer types. Another example regarded the breast invasive carcinoma (BRCA) patients, which consisted of 14 basal and 61 non-basal cases. These two subtypes of BRCA corresponded to two patient clusters that were clearly separated from one another (Additional file 1: Fig. S11).

Next, we focused on the four super classes of cancer types (i.e., kidney carcinoma, brain cancer, SC, and DIAD) and explored the common properties of each class from the perspective of transcriptional regulators. Technically, we obtained 521 binding motifs of 432 different human TFs from the JASPAR database [45] and quantitatively inferred the activity of the TFs in each ATAC-seq sample (an individual TF could be associated with multiple motif versions). To achieve that, we first identified the instances of each motif in the genome by employing a motif-scanning routine [46]. Then, for each motif, we took the HVRs containing its instances and used the average ATAC-seq signal across these HVRs (in each sample) as an activity score of the corresponding TF (the ATAC-seq signals in different HVRs had been separately scaled beforehand; see the "Methods" section). Finally, we identified class-specific TFs by comparing the activity scores derived from each motif between the samples belonging to each class and the other samples, with the application of $t$-tests (Additional file 4: Table S4). Figure 6e illustrates several representative (top-ranked) motifs for each of the four classes.

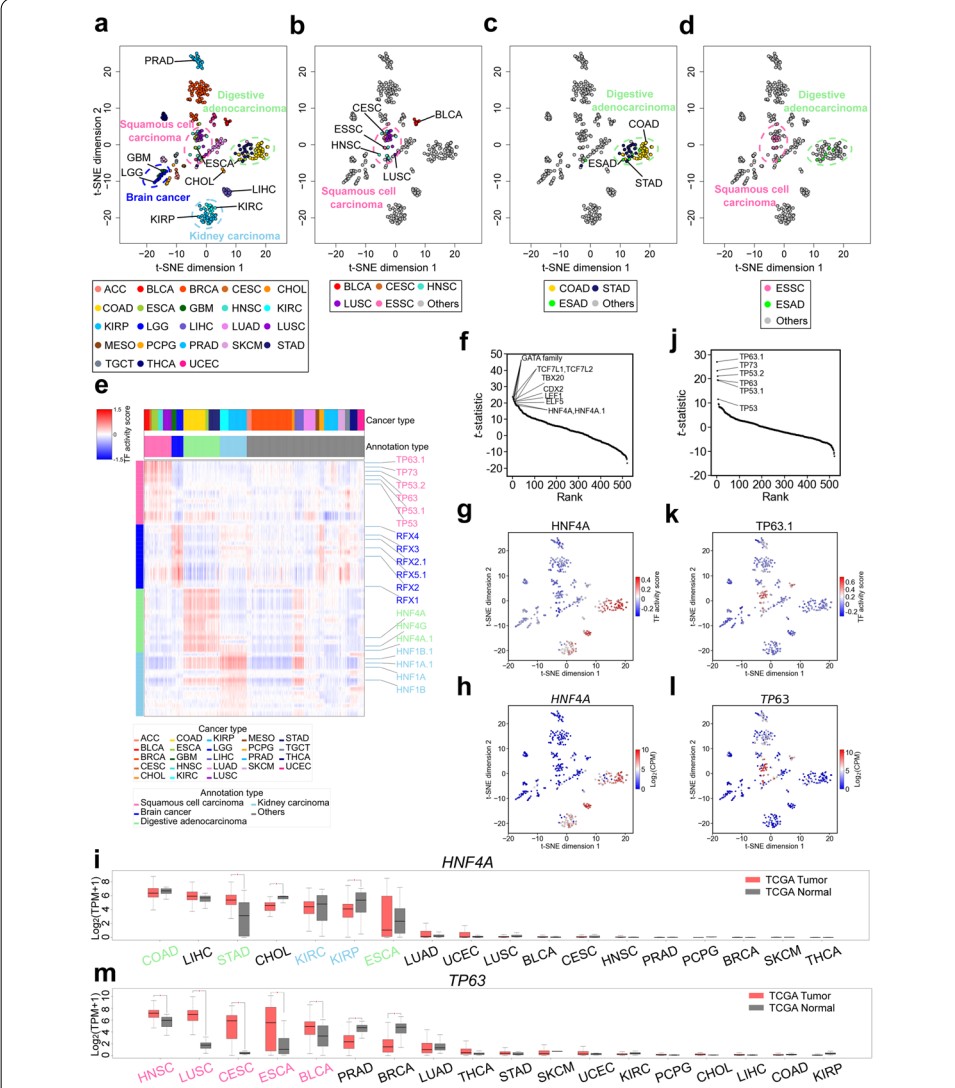

**Fig. 6** Applying HyperChIP to a TCGA pan-cancer ATAC-seq data set. **a** Two-dimensional t-SNE plot of all patients based on their ATAC-seq profiles of tumor tissues. The patients are colored by cancer types (see Additional file 3: Table S3 for the full names of the involved cancer types). **b** The distribution of the patients belonging to the SC class. **c** The distribution of the patients belonging to the DIAD class. **d** The distribution of all ESCA patients, comprising 12 ESSC and 6 ESAD cases. **e** Heat map showing the TF activity scores of the top 15 binding motifs identified for each class of cancer types. **f** Plotting the *t*-statistics of all motifs against their rankings in the identification of DIAD class-specific TFs. **g** Mapping the TF activity scores of the HNF4A motif to the t-SNE plot. **h** Mapping the expression levels of the *HNF4A* gene (calculated from the corresponding RNA-seq samples) to the t-SNE plot. **i** Box plots showing the expression of *HNF4A* in a larger TCGA cohort of (7183) patients. The expression data are accessed via the online tool GEPIA, in which ESSC and ESAD patients are both labeled ESCA and cannot be distinguished from each other. The cancer types are sorted by the median expression of *HNF4A* in tumors. For each cancer type, the significance of differential expression is determined by performing a *t*-test with a *p*-value cutoff of 0.01 and a fold change cutoff of 2. TPM, transcripts per million. **j** Plotting the *t*-statistics of all motifs against their rankings in the identification of SC class-specific TFs. **k** Mapping the TF activity scores of the TP63.1 motif to the t-SNE plot. **l** Mapping the expression levels of the *TP63* gene to the t-SNE plot. **m** Box plots showing the expression of *TP63* in the larger TCGA cohort

We noticed that a number of the identified class-specific TFs have been reported to have tissue-specific activity in the corresponding organs. For example, *HNF4A* is an important TF linked with the regulation of liver-specific gene expression as well as

multiple biological processes in the epithelia of the gastrointestinal tract and kidneys [47, 48]. In our analysis, both HNF4A and HNF4A.1 were identified as top-ranked motifs for the DIAD class (Fig. 6f), of which all the three cancer types originated in organs of the gastrointestinal tract (i.e., the esophagus, stomach, and large intestine). Compared to the other cancer types, ATAC-seq samples of the DIAD and kidney carcinoma classes as well as those of the liver-associated cancer types, including liver hepatocellular carcinoma (LIHC) and cholangiocarcinoma (CHOL), showed clearly higher HNF4A activity scores (Fig. 6g). Consistently, based on the matched RNA-seq samples, we found *HNF4A* was expressed almost exclusively in these cancer types (Fig. 6h). We further accessed the expression strength of *HNF4A* in tumor tissues and adjacent normal tissues of a much larger cohort of cancer patients, by using the GEPIA web server [49]. A consistent tissue specificity of *HNF4A* was observed in this cohort as well, and we found no systematic differential expression of *HNF4A* between the tumor and matched normal tissues (Fig. 6i). Another example was *HNF1A*, a TF specifically expressed in organs of endoderm origin, including the kidneys and almost all the digestive organs [50]. In our analysis, *HNF1A* was identified as a top-ranked TF for the kidney carcinoma class, and both of its activity scores and expression strength showed consistent tissue specificity with its biological roles (Additional file 1: Fig. S12a-c). Again, we found no systematic differential expression of *HNF1A* between the tumor and matched normal tissues (Additional file 1: Fig. S12d).

By contrast, multiple top-ranked TFs identified for the SC class were common oncogenes shared by the involved SC types. For example, *TP63* has been implicated as an oncogene in several SC types, including head and neck squamous cell carcinoma (HNSC) [51], LUSC [52], and ESSC [53]. In our analysis, *TP63* was the top 1 TF for the SC class (Fig. 6j). Compared to the other cancer types, the cancer types of the SC class exhibited considerably higher activity scores and expression levels of *TP63* (Fig. 6k, l). We next examined the expression of *TP63* in the larger cohort of cancer patients. Notably, for each SC type, *TP63* showed significantly higher expression in the tumor tissues than in the matched normal tissues, and such upregulation of *TP63* was not observed in any non-SC cancer types (Fig. 6m). These results suggested *TP63* could be a "pan-SC" oncogenic TF. Another example was *TP73*, which ranked second for the SC class. Similar to *TP63*, *TP73* showed systematically increased expression in the tumor tissues of the SC types (Additional file 1: Fig. S13).

There was also a third type of class-specific TFs, which was associated with strong tissue specificity as well as significantly differential expression between tumor and matched normal tissues. For example, *RFX4* is a TF specifically expressed in the brain [54], and it has been recently revealed to be associated with tumor progression in patients with glioblastoma [55]. In our analysis, *RFX4* ranked first for the brain cancer class (Additional file 1: Fig. S14a). Among all the cancer types, *RFX4* was exclusively expressed in GBM and LGG, and its expression in them was significantly higher than in the matched normal tissues (Additional file 1: Fig. S14b-d). Together, these results indicated the HVRs identified by HyperChIP can contribute to the identification of regulators pertaining to the heterogeneity across samples.

### Applying HyperChIP to non-cancer data sets

Chromatin states vary extensively even across normal humans [28, 42]. To explore the usefulness of hypervariable ChIP/ATAC-seq analysis in this context, we applied

HyperChIP to a CTCF ChIP-seq data set comprising samples of 17 lymphoblastoid cell lines (LCLs) derived from different human individuals, including 6 Caucasian individuals (GM10847, GM12878, GM12890, GM12891, GM12892, SNYDER), 7 Yoruban individuals (GM18486, GM18505, GM19099, GM19193, GM19238, GM19239, GM19240), and 4 individuals from the San population (GM2255, GM2588, GM2610, GM2630) [28]. In total, 364 proximal HVRs and 498 distal ones were identified (BH-adjusted *p*-value < 0.1). We then performed principal component analysis (PCA) of the CTCF ChIP-seq samples with all peak regions or only the HVRs as features. Interestingly, it was found that only in the latter case were the LCLs well clustered by their populations of origin (Fig. 7a, b). This finding suggested that the hypervariable CTCF binding signals captured by HyperChIP across the LCLs were useful for dissecting the similarity structure among them.

To illustrate the utility of HyperChIP in analyzing ChIP/ATAC-seq samples from time course experiments, we further incorporated a mouse ATAC-seq data set that profiled the chromatin accessibility in preimplantation embryos at different development stages [56]. In detail, this data set contained 2 biological replicates separately for the 2-cell, 4-cell, and 8-cell embryos and the inner cell masses (ICMs) of the blastocysts, as well as 3 biological replicates for mouse embryonic stem cells (mESCs; derived from ICMs). Applying HyperChIP, we identified 303 proximal HVRs and 383 distal ones (BH-adjusted *p*-value < 0.1). PCA with these HVRs as features revealed that a large proportion (71.6%) of the ATAC-seq signal variability at these regions was accounted for by the first principal component, which showed a strong association with the development timeline (Fig. 7c). We then accordingly classified the samples into early-stage (2-cell, 4-cell, and 8-cell embryos) and late-stage (ICMs and mESCs) ones, and the same motif analysis as applied to the pan-cancer ATAC-seq data set was repeated to identify stage-specific regulators (Fig. 7c, d; Additional file 5: Table S5). Consistent with the findings in the original study [56], both *Rarg* and *Nr5a2* were identified as top-ranked TFs associated with the early stage, and their expression showed a clear and consistent difference between the two stages as well (Fig. 7e; Additional file 1: Fig. S15a). Another example was the *Obox* family, which is a homeobox gene family and has been implicated in early embryonic development [57, 58]. In particular, it has been found that the transcripts of both *Obox2* and *Obox3* are most abundant in the 1-cell embryos with the abundance decreasing in further development until no expression is observed in the morulae (before the blastocysts), and the expression of *Obox6* is concentrated between the 2-cell embryos and the morulae. For the late stage, both *Sox2* and *Klf4* (known regulators of ICMs and mESCs) were identified as top-ranked TFs (Fig. 7f). Another example was *Gbx2* (Additional file 1: Fig. S15b), which is an upstream regulator of *Klf4* and maintains naïve pluripotency of mESCs by inducing the expression of *Klf4* [59]. Notably, many of the top-ranked TFs associated with the early/late stage showed stepwise decreased/increased activity scores along the development timeline, such as *Obox3* and *Klf4* (Fig. 7d–f). Together, these observations suggest that applying HyperChIP to samples from a time course experiment contributes to revealing regulators pertaining to the dynamics during the biological process.

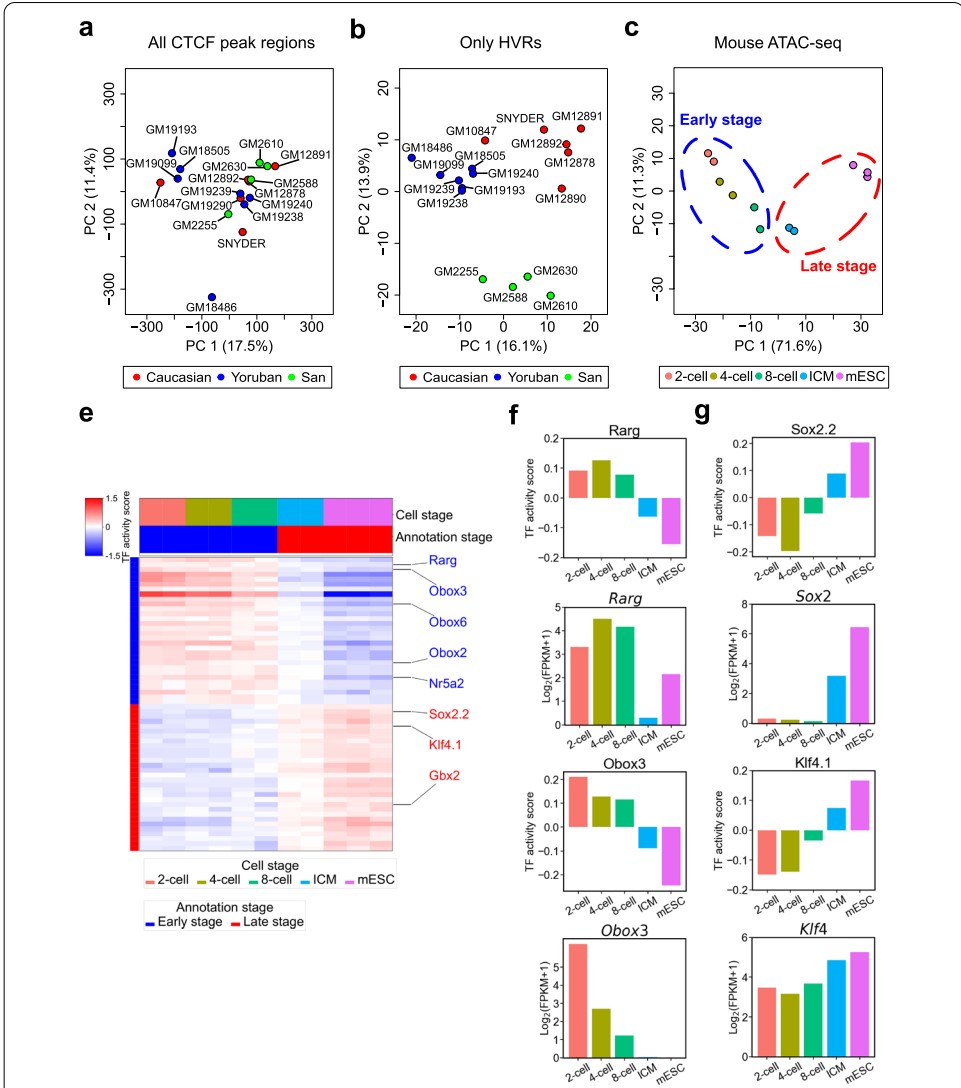

**Fig. 7** Applying HyperChIP to non-cancer ChIP/ATAC-seq data sets. **a**, **b** PCA results for the CTCF ChIP-seq data set, which comprised samples of 17 human LCLs with different populations of origin. PCA is performed either **a** with all peak regions or **b** with only the identified HVRs as features. **c** PCA results for the mouse ATAC-seq data set, which comprised samples of preimplantation embryos at different development stages and mESCs. Only the identified HVRs are used as features. **d** Heat map showing the TF activity scores of the top 30 binding motifs identified for the early/late-stage defined in **c**. **e** Bar plots showing the TF activity scores and expression levels of *Rarg* and *Obox3*, both of which are top-ranked TFs associated with the early stage. TF activity scores have been averaged across biological replicates for each individual cell stage. FPKM, fragments per kilobase of transcript per million fragments mapped. **f** Bar plots showing the TF activity scores and expression levels of *Sox2* and *Klf4*, both of which are top-ranked TFs associated with the late stage

## Discussion

Hypervariable ChIP/ATAC-seq analysis plays an essential role in large-scale epigenetic studies, considering its importance in dissecting the similarity structure among samples. In the study, we presented HyperChIP as the first complete statistical tool for the computational task. HyperChIP accounts for the mean-variability dependence intrinsic to count data by fitting an MVC, and it increases the statistical power by selecting a subset

of genomic regions with a low abundance of HVRs and LVRs and employing winsorization in the parameter estimation procedure.

The specific hypervariable analysis method directly affects the reliability of various downstream analyses, including the interpretation of HVRs as well as the dissection of the similarity structure among samples. For the LUAD H3K27ac ChIP-seq data set, we found that the HVRs identified by HyperChIP were strongly associated with the different tumor progression stages of patients (Fig. 5f, g). Here, we applied each of the methods that were used in benchmarking HyperChIP to this analysis (the same number of top-ranked proximal/distal HVRs as identified by HyperChIP was selected for each method). For these methods, the correlations of selected HVRs with tumor progression stage were either roughly as strong as observed from HyperChIP or weaker (Additional file 1: Fig. S16). We also applied these methods to the pan-cancer ATAC-seq data set and repeated the t-SNE analysis as presented in Fig. 6a. On the one hand, the two-dimensional t-SNE plots generated by different methods were similar to each other (Additional file 1: Fig. S17a-g). Based on the HVRs derived by each method (more precisely, the same principal components as used in the t-SNE analysis), we performed a classification of all samples and assessed their agreement with the cancer type labels by calculating the ARI. The ARI values achieved by different methods were close to each other as well (Additional file 1: Fig. S17h). On the other hand, HyperChIP showed better performance in revealing fine structures among the samples. For example, the kidney carcinoma class consisted of KIRC and KIRP samples. These samples were very close together in all the t-SNE plots, but only in the t-SNE plot generated by HyperChIP were the two cancer types clearly separated from each other (Additional file 1: Fig. S18a-g). Moreover, we performed classifications specifically for the KIRC and KIRP samples (the same features were used as in the previous classifications of all samples) and found that HyperChIP achieved the highest ARI in separating the two cancer types (Additional file 1: Fig. S18h). Similar results were also observed between the GBM and LGG types in the brain cancer class as well as between the colon adenocarcinoma (COAD) and stomach adenocarcinoma (STAD) types in the DIAD class (Additional file 1: Figs. S19, S20).

Throughout the entire study, we have strictly followed the criterion of separately handling proximal and distal regions in all hypervariable ChIP/ATAC-seq analyses. In summary, the variability structure associated with proximal regions differs from that of distal regions in many respects, including the underlying mean-variance trend, the distribution of HVRs along the range of mean signal intensities, and the global variability magnitude. Regarding the mean-variance trend, we have seen that the MVCs fitted for each of the data sets in Table 1 are distinct between proximal and distal regions (Additional file 1: Fig. S1). If we do not separate proximal and distal regions when applying HyperChIP, the MVC fitting procedure will have to compromise between the two classes of regions, which will influence not only the overall rankings of all peak regions but also the rankings within each class. Here, we tried applying HyperChIP to the data sets in Table 1 without separately handling proximal and distal regions. It was found that the rankings of proximal/distal regions became worse with respect to the consistency with HVGs and the resulting classifications of samples (Additional file 2: Note S2.1). We also examined the overall rankings of all peak regions, and we still observed that separating proximal and distal regions can bring an improvement in the performance of HyperChIP

(in which case the two classes of regions were ranked together based on their respective *p*-values or BH-adjusted *p*-values; Additional file 2: Note S2.2).

Regarding the global variability magnitude, the ChIP/ATAC-seq signal variability in distal regions is typically higher than that in proximal regions. Here, to provide a clear demonstration of this difference, we design a pipeline for comparing the global signal variability (across a given set of ChIP/ATAC-seq samples) between proximal and distal regions. Technically, we achieve it by comparing the variance ratio factors (denoted by $\gamma$) separately estimated from proximal and distal regions, while controlling for the MVC and the $d_0$ parameter (refer to the HyperChIP model formulated by Eqs. (1) and (2)). In detail, we first process together proximal and distal regions in the normalization and MVC fitting procedures. Then, we set $d_0$ to positive infinity and apply the winsorization framework (see the "Methods" section) separately to proximal and distal regions to estimate $\gamma$ (note that no selection of low-intensity regions is made). In this way, the $\gamma$ estimate derived from proximal/distal regions represents the scaling factor of the MVC to reach the variance values observed at proximal/distal regions, and we can effectively compare the global signal variability between proximal and distal regions by comparing the corresponding $\gamma$ estimates. Applying this pipeline, we found that the $\gamma$ estimate derived from proximal regions was smaller than that from distal regions for each of the data sets in Table 1 (Fig. 8a). We further split the pan-cancer ATAC-seq data set into 23 small data sets corresponding to individual cancer types. Again, we observed a systematic increase in signal variability at distal regions compared to proximal regions (Fig. 8b). Together, these observations indicated the necessity of separately handling proximal and distal regions in hypervariable ChIP/ATAC-seq analysis.

## Conclusions

HyperChIP has been presented as a complete statistical tool for identifying HVRs on ChIP/ATAC-seq samples. It uses local regression to adaptively capture the mean-variance relationship, which leads to better statistics for ranking HVRs compared to other existing ones. Specific efforts have been made to alleviate the influence of HVRs and LVRs on model fitting, which effectively increases the statistical power of HyperChIP. Case studies indicate that the HVRs identified by HyperChIP not only provide a solid basis to uncover the similarity structure among the involved samples, but can also contribute to the identification of regulators pertaining to the similarity structure when coupled with a motif-scanning procedure.

## Methods

### Data preprocessing

Processing of the mouse ATAC-seq samples, the RNA-seq samples associated with the data sets in Table 1, and all the ChIP-seq samples used in the study started with raw sequencing reads. We first used Trim Galore (v0.4.4) to trim 3′ ends of reads [60]. Resulting RNA-seq and ChIP/ATAC-seq reads were then aligned to the hg19/mm10 reference genome by STAR (v2.5.1b) and Bowtie (v1.2.2), respectively [61, 62]. To avoid artifacts from PCR amplification, we kept for each sample at most one read or read pair at each genomic location. The remaining reads or read pairs of each RNA-seq sample were then assigned to UCSC annotated genes [63] by htseq-count (v0.6.1p1) [64]. For the NSCLC

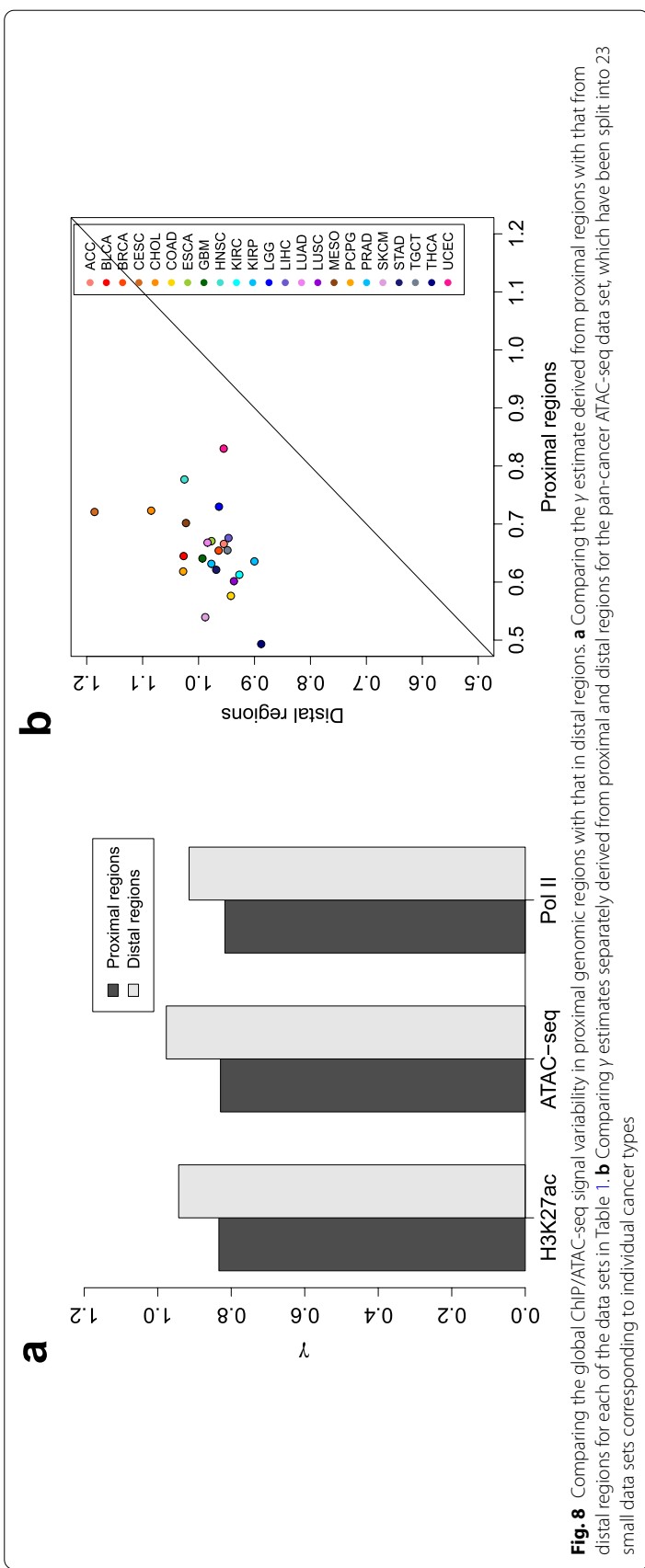

**Fig. 8** Comparing the global ChIP/ATAC-seq signal variability in proximal genomic regions with that in distal regions. **a** Comparing the γ estimate derived from proximal regions with that from distal regions for each of the data sets in Table 1. **b** Comparing γ estimates separately derived from proximal and distal regions for the pan-cancer ATAC-seq data set, which have been split into 23 small data sets corresponding to individual cancer types

ATAC-seq samples, read alignments to hg19 (BAM-formatted files) were directly obtained from the corresponding study. Trimming of reads and removal of duplicates had already been done for these BAM files. For the pan-cancer ATAC-seq data set, a count matrix recording the read counts for each sample at a list of genomic regions was downloaded from the TCGA page https://gdc.cancer.gov/about-data/publications/ATACseq-AWG.

For the pan-cancer and mouse ATAC-seq data sets, the associated gene expression values, which were respectively CPM and FPKM values derived from RNA-seq data, were directly obtained from the corresponding studies. Accompanying input samples measuring background signals were available for all the ChIP-seq samples and were processed in the same way as were the ChIP-seq samples.

### Processing read alignments and calling peaks

For paired-end ChIP-seq and input samples, we converted each read pair into a single read whose 5′ end lay upstream of the associated DNA fragment center by 100 bp, with the fragment center inferred as the midpoint between the two 5′ ends of the read pair. For ATAC-seq read alignments, we shifted upstream each individual read by 100 bp. The whole process was for making the 5′ ends of all the resulting reads lie upstream of the presumed protein binding sites by a fixed distance.

Peak calling for each ChIP-seq sample was then performed against the corresponding input sample by using MACS (v1.4.2), with the parameters "--nomodel --shiftsize=100 --keep-dup=all" [65]. For the NSCLC ATAC-seq samples, peaks were directly obtained from the corresponding study. For the mouse ATAC-seq samples, we used MACS (v1.4.2) with the same parameters, except that no input samples were provided. We also assessed the quality of each data set based on the distributions of peak numbers and fractions of reads in peaks (FRiPs) (see Additional file 2: Note S3 for details).

### Input matrices for normalization

A count matrix and an occupancy matrix have been constructed for each data set. Rows and columns of both matrices corresponded to a list of genomic regions and the related ChIP/ATAC-seq samples, respectively. The count matrix recorded raw read counts. The occupancy matrix used binary variables to indicate whether each region was a peak region in each sample.

For all the data sets but the pan-cancer ATAC-seq one, the count and occupancy matrices were generated by invoking MAnorm2_utils (v1.0.0) [24] with the parameters "--typical-bin-size=X --shiftsize=100 --keep-dup=all --filter=blacklist" (X was set to 2000 and 1000 for the H3K27ac ChIP-seq data set and the other data sets, respectively). The blacklisted regions for hg19/mm10 were obtained from Amemiya et al. [66]. For the pan-cancer ATAC-seq data set, the occupancy matrix was determined by considering the regions with > 50 read counts as peak regions.

For the two matrices associated with each data set, we then removed the genomic regions on sex chromosomes to avoid the influence of different biological sexes. We also classified the remaining regions into proximal and distal ones and accordingly split each matrix. Technically, each region with a distance less than 5 kb to any (UCSC annotated) transcription start site was considered proximal, and it was considered distal otherwise (see Additional file 2: Note S4 for a detailed discussion of this distance cutoff). Unless

otherwise stated, each normalization/hypervariable analysis targeting an individual data set has separately handled proximal and distal regions.

### Normalization

We detail here the method used in the study for deriving normalized $\log_2$ read counts as input data of HyperChIP. Given a count matrix and an occupancy matrix, suppose $K_{ij}$ is the read count at region $i$ in sample $j$ and that $O_{ij}$ indicates the associated occupancy status. Each $O_{ij}$ takes a value of 0 or 1, and $O_{ij}=1$ indicates region $i$ is a peak region in sample $j$ (or region $i$ is occupied by sample $j$). Define $Y_{ij}=\log_2(K_{ij}+0.5)$.

To normalize the samples, we first construct a pseudo-reference profile as normalization baseline, which is defined by:

$$P_i = \begin{cases} \frac{\sum_j O_{ij} Y_{ij}}{\sum_j O_{ij}} & \sum_j O_{ij} > 0, \\ 0 & \sum_j O_{ij} = 0. \end{cases} \tag{4}$$

The above definition is for notational convenience. In fact, the $P_i$ with which the associated $\sum_j O_{ij}$ are 0 will never be used in the subsequent normalization procedures.

We next repeatedly normalize each sample against the baseline. Technically, the normalization of sample $j$ is accomplished by applying a linear transformation to all the corresponding $Y_{ij}$. Formally, let $X_{ij}=\alpha_j+\beta_j Y_{ij}$ be normalized $\log_2$ read counts, where $\alpha_j$ and $\beta_j$ are coefficients to be determined. For notational simplicity, we further define $M$ and $A$ values as $M_{ij}=X_{ij}-P_i$ and $A_{ij}=(P_i+X_{ij})/2$, respectively, and define $n_j=\sum_i O_{ij}$ to be the total number of regions that are occupied by sample $j$. The two coefficients are determined by imposing the following two constraints:

$$\sum_{i:O_{ij}=1} M_{ij} = 0, \tag{5}$$

$$\sum_{i:O_{ij}=1} \left( M_{ij} - \frac{\sum_{i':O_{i'j}=1} M_{i'j}}{n_j} \right) \left( A_{ij} - \frac{\sum_{i':O_{i'j}=1} A_{i'j}}{n_j} \right) = 0. \tag{6}$$

Intrinsically, the above two constraints are for simultaneously removing the global signal difference and *M-A* trend at common peak regions of sample $j$ and the pseudo-reference profile. Solutions for $\alpha_j$ and $\beta_j$ are given by:

$$\alpha_j = \frac{\sum_{i:O_{ij}=1} P_i - \beta_j \sum_{i:O_{ij}=1} Y_{ij}}{n_j}, \tag{7}$$

$$\beta_j = \sqrt{\frac{\sum_{i:O_{ij}=1} \left( P_i - \frac{\sum_{i':O_{i'j}=1} P_{i'}}{n_j} \right)^2}{\sum_{i:O_{ij}=1} \left( Y_{ij} - \frac{\sum_{i':O_{i'j}=1} Y_{i'j}}{n_j} \right)^2}}. \tag{8}$$

**Parameter estimation for the HyperChIP model**

The whole model formulation of HyperChIP as well as the associated hypothesis testing procedure is given by the Eqs. (1), (2), and (3). Here, we detail the method for estimating $d_0$ and $\gamma$.

We first explain the application of the winsorization technique. The idea is similar to robust limma [67], and we apply moment estimation to winsorized log-scaled variances. Formally, let $z_i = \log \frac{\hat{t}_i}{f(\hat{\mu}_i)}$ be the log-scaled variance associated with region $i$. Set $p_l$ and $p_u$ to two small values representing the maximum proportions of outliers allowed in the lower and upper tails of all $z_i$, respectively (in the study, $p_l = 0.01$ and $p_u = 0.1$ were always used). Let $q_l$ and $q_u$ be the corresponding lower and upper sample quantiles of $z_i$, respectively. Then, winsorized $z_i$ are defined by:

$$\text{win}(z_i; p_l, p_u) = \begin{cases} q_l & z_i \leq q_l, \\ z_i & q_l < z_i < q_u, \\ q_u & z_i \geq q_u. \end{cases} \tag{9}$$

Based on the model, it approximately follows:

$$\text{win}(z_i; p_l, p_u) \sim \log \gamma + \log \left[ \text{win} \left( F_{m-1, d_0}; p_l, p_u \right) \right]. \tag{10}$$

Note that, in contrast with the winsorization of $z_i$, the winsorization of $F_{m-1, d_0}$ in the above formula refers to squeezing certain proportions at two tails of the probability density function towards the corresponding theoretical quantiles, which results in a mixture of a continuous distribution with a bounded support domain and two point masses at the edges.

We next apply a moment estimation approach. It follows from Eq. (10):

$$E[\text{win}(z_i; p_l, p_u)] = \log \gamma + E\left\{ \log \left[ \text{win} \left( F_{m-1, d_0}; p_l, p_u \right) \right] \right\}, \tag{11}$$

$$\text{var}\left[\text{win}(z_i; p_l, p_u)\right] = \text{var}\left\{ \log \left[ \text{win} \left( F_{m-1, d_0}; p_l, p_u \right) \right] \right\}. \tag{12}$$

The approach is to first solve Eq. (12) for $d_0$ and then solve Eq. (11) for $\gamma$. Technically, given $d_0$, the expectation and variance on the right-hand sides of (Eqs. (11) and (12)) , respectively, are calculated by turning to Gauss-Legendre quadrature [68], and Eq. (12) is solved by applying the bisection method.

Besides using winsorization, HyperChIP also makes a selection of low-intensity regions for parameter estimation. By default, the above winsorization process is only applied to the $z_i$ of the 10% of regions with the smallest $\hat{\mu}_i$.

**Applying other methods for ranking HVRs**

For the methods that do not consider the mean-variability dependence, the associated variability statistics, including observed variance, MAD, and IQR, were calculated from the same normalized $\log_2$ read counts as used by HyperChIP. For the min-rank method, it was applied to the same normalized data as well. For the method that uses a moderately large offset when applying a $\log_2$ transformation, we followed the computational pipeline presented in the original study [17]. More specifically, we first derived $\log_2$-CPM

values by using the cpm function of the edgeR package [69], with setting log=TRUE and prior.count to a large value (5 or 10 in this study). We then applied quantile normalization to the $\log_2$-CPM values and calculated the observed variance associated with each region. Finally, these variances were used to rank HVRs.

### Identifying HVGs

We identified HVGs separately for each data set in Table 1, by applying limma trend [23, 70] to the corresponding RNA-seq samples. Technically, we first converted RNA-seq read counts into $\log_2$-CPM values by using the calcNormFactors and cpm functions of the edgeR package, with log=TRUE for the latter. Then, the standard limma pipeline [71] was applied to the $\log_2$-CPM values, with trend=TRUE for the eBayes function (note that the design matrix contained only an intercept variable). The ratios of sample variances to prior variances were then used as key statistics for identifying HVGs, and we applied a one-sided *p*-value cutoff of 0.05.

### Classification of the NSCLC ATAC-seq samples

Each classification of the samples was performed by using a set of top-ranked proximal/distal HVRs derived by some method as features. For all the methods but the large-offset one, we first separately scaled the normalized $\log_2$ read counts associated with each feature. For the large-offset method, the normalized data bound with it were directly used for the subsequent clustering analysis [17]. Then, we calculated the Euclidean distance between each pair of samples and performed a hierarchical clustering by using the hclust function of the R software [72], with method="ward.D". Finally, the resulting hierarchical tree was cut into two branches, and we assessed the agreement of this classification with the subtype labels (LUAD or LUSC) by calculating the ARI, which was achieved by using the adjustedRandIndex function of the mclust package [73].

### Biological interpretation of the significant HVRs and LVRs identified for the LUAD H3K27ac ChIP-seq data set

For the GO enrichment analysis, GO terms for biological processes were obtained from the Molecular Signatures Database (MSigDB) [74]. Enrichments of GO terms were assessed by performing Fisher's exact tests, with all the proximal H3K27ac peak region-associated genes as background.

For the histopathological labels of the LUAD patients, we first made a classification based on the most prevalent histologic pattern of each patient, which was lepidic, papillary, acinar, or solid [75]. According to the original study [6], we then specified the histopathological labels of the lepidic-prevalent and solid-prevalent patients as low-risk and high-risk, respectively, and the remaining patients were considered median-risk. Finally, we ranked all patients based on their risk levels and quantified the Spearman correlations of H3K27ac ChIP-seq signals in different regions with the risk levels.

We also separately performed a regression of the survival time of the patients on the H3K27ac ChIP-seq signals in each region, by fitting a Cox proportional hazards model [41]. Technically, this was achieved by using the CoxPHFitter function of the lifelines package [76], and we used the (two-sided) *p*-value associated with each model to evaluate the prognostic association of the corresponding region. For the hierarchical clustering of

the patients, we followed the same pipeline as used for classifying the NSCLC ATAC-seq samples, except that the proximal and distal HVRs were used together as features. Testing the survival difference between the resulting two subgroups was achieved by using the survdiff function of the survival package [77].

The phastCons scores for assessing the sequence conservation of different regions were obtained from the phastCons100way.UCSC.hg19 package [36]. Each base with a phastCons score over 0.9 was considered conserved.

## Identifying ATAC-seq QTLs

For the NSCLC ATAC-seq data set, among the germline SNPs located within the peak regions, we have identified QTLs whose different genotypes were associated with significantly differential ATAC-seq signals in the enclosing peak region. For this analysis, we first noticed that all the SNPs were associated with exactly two different genotypes across the individuals, referred to as reference and alternative genotypes. To increase the statistical power, we filtered out the SNPs whose reference or alternative genotype was associated with less than 5 individuals. Then, for each remaining SNP, we performed a two-sample *t*-test between the ATAC-seq signals (i.e., the normalized $\log_2$ read counts used by HyperChIP) associated with the two genotypes. Finally, the SNPs with a BH-adjusted *p*-value less than 0.1 were considered as significant QTLs.

## Analysis of the pan-cancer ATAC-seq data set

For the t-SNE analysis, we first separately scaled the ATAC-seq signals at each significant HVR. Formally, suppose $X_{ij}$ is the normalized $\log_2$ read count at HVR $i$ in sample $j$. Let $X_{i\cdot} = (X_{i1}, X_{i2}, \cdots)$ be the vector of normalized $\log_2$ read counts at HVR $i$ in all the samples. We define:

$$Z_{ij} = \frac{X_{ij} - \mathrm{mean}(X_{i\cdot})}{\mathrm{sd}(X_{i\cdot})}, \tag{13}$$

where mean and sd refer to the sample mean and sample standard deviation, respectively. Let $Z_{\cdot j} = (Z_{1j}, Z_{2j}, \cdots)$ be the scaled ATAC-seq signals at all the HVRs in sample $j$. We then performed a PCA by using $Z_{\cdot j}$ as the features of sample $j$. Finally, the t-SNE routine implemented in the Rtsne package [78] was applied to the first 50 principal components, which led to the two-dimensional t-SNE plot.

For the identification of class-specific regulators, we first performed a motif-scanning analysis by using the matchMotifs function of the motifmatchr package [46] with default settings. Let $I_{mi}$ indicate whether HVR $i$ contains an instance of motif $m$. Each $I_{mi}$ takes a value of 0 or 1. Then, we defined the activity score of each motif $m$ in each sample $j$ as:

$$S_{mj} = \frac{\sum_i I_{mi}\left(Z_{ij} - \mathrm{mean}\left(Z_{\cdot j}\right)\right)}{\sum_i I_{mi}}, \tag{14}$$

where the sum operators are applied to all the HVRs and the subtraction is for adjusting for sample-specific biases. Finally, for each of the four classes of cancer types, we performed a two-sample *t*-test for each motif to compare its activity scores between the samples belonging to the class and the other samples.

### Analysis of the non-cancer data sets

For the CTCF ChIP-seq data set of human LCLs as well as the ATAC-seq data set of mouse preimplantation embryos and mESCs, PCA was performed based on the same procedure as used for the pan-cancer ATAC-seq data set. For the mouse ATAC-seq data set, the identification of early/late stage-specific regulators was achieved by following the same pipeline as applied to the pan-cancer ATAC-seq data set, except that binding motifs of mouse TFs instead of human ones were used.

### Supplementary Information

**Additional file 1: Figure S1.** Scatter plots showing various mean-variance trends associated with different data sets. Variance is shown at the log10 scale. Red lines depict the corresponding MVCs. Red points mark the 1000 regions with the largest scaled variances. **Figure S2.** Scatter plots of log10 scaled variances against observed mean signal intensities for different data sets. Red points mark the 1000 regions with the largest scaled variances. **Figure S3.** Applying other methods for ranking genomic regions and selecting HVRs. (a-c) Scatter plots showing the mean-variance trend (at proximal regions) associated with the H3K27ac ChIP-seq data set as well as the regions that are ranked in the top 1000 HVRs by each method (marked by red points). MAD, median absolute deviation; IQR, interquartile range. **Figure S4.** Identifying HVGs. We have separately identified HVGs for each data set in Table 1, by applying limma-trend to the corresponding RNA-seq data (see Methods in the main text for details). Scatter plots shown here demonstrate the modeling of the mean-variance relationships by limma-trend. Red points in each plot mark the identified HVGs. CPM, count per million; SD, standard deviation. **Figure S5.** Selecting a subset of genomic regions and using Winsorization for parameter estimation. (a) For the ATAC-seq and Pol II ChIP-seq data sets, bar plots showing the distributions of top-ranked proximal/distal HVRs along the range of mean intensities. For each data set, proximal and distal regions have been separately divided into 10 equally-sized groups based on the observed mean signal intensities. (b) Bar plots showing the distributions of top-ranked proximal/distal LVRs along the range of mean intensities. (c) $d0$ estimates resulting from different parameter estimation methods. Inf refers to positive infinity. **Figure S6.** The distributions of top-ranked proximal HVRs and LVRs associated with the Pol II ChIP-seq data set. The top-ranked LVRs form two clusters that are somewhat separated from one another in the mean-variance scatter plot, owing to a gap (indicated by the area circled in green) largely corresponding to the 70th to 90th percentile of mean intensities. As a result, the proportion of the LVRs dips at the corresponding two groups of regions, leading to a bimodal distribution profile as well as a rise in the HVR proportion that is more dramatic compared to the other two data sets. **Figure S7.** Evaluating the prognostic associations of different genomic regions. (a, b) Proximal/distal HVRs are more significantly associated with the survival time of patients than proximal/distal LVRs and randomly selected proximal/distal peak regions. Results shown here are based on the H3K27ac ChIP-seq data set. The *p*-values are derived by separately performing a Cox regression on the H3K27ac level in each region. (c) Dendrogram showing the hierarchical clustering of the patients based on the proximal and distal HVRs. The patients are classified into two sub-groups, labeled C1 and C2. (d) There is a significant survival difference between C1 and C2. **Figure S8.** HVRs have a significant association with somatic copy number variation (CNV). (a, b) Proximal/distal HVRs are more likely to be associated with somatic CNV than randomly selected proximal/distal peak regions. Results shown here are based on the ATAC-seq data set. For this data set, the genomic segments with somatic CNV in at least one patient together occupied almost the whole genome (>95%). We therefore considered a region as associated with somatic CNV only if it overlapped CNV segments in more than 5 patients. 1,000 random simulations were performed separately for proximal and distal peak regions. In each time, we randomly selected the same number of proximal/distal peak regions as that of the proximal/distal HVRs. **Figure S9.** HVRs contain significantly more germline SNPs than by chance. (a) Proximal HVRs identified for the ATAC-seq data set are enriched with germline SNPs. We have performed 1,000 times of random simulation. In each time, a set of proximal peak regions matching the number of the HVRs has been randomly selected. (b) Distal HVRs are enriched with germline SNPs as well. **Figure S10.** Association between QTLs and HVRs. (a, b) Identifying QTLs among the germline SNPs located within ATAC-seq peak regions. Each (BH-adjusted) *p*-value assesses the statistical significance of the association between the genotype of a SNP and the ATAC-seq signal in the peak region containing it. (c) Box plots showing the ATAC-seq signals associated with different genotypes of the most significant QTL, which is located within a proximal HVR. Ref and Alt refer to the reference genotype and the alternative one, respectively. (d) Box plots showing the RNA-seq signals of the downstream gene of the proximal HVR. (e) Box plots showing the ATAC-seq signals associated with different genotypes of the most significant distal QTL, which is also located within an HVR. (f) Box plots showing the RNA-seq signals of the gene nearest to the distal HVR. **Figure S11.** For the TCGA pan-cancer ATAC-seq data set, two-dimensional t-SNE plots showing the distribution of BRCA patients. These patients are comprised of 14 basal and 61 non-basal cases. **Figure S12.** *HNF1A* is identified as a top-ranked TF for the kidney carcinoma class. (a) Plotting the *t*-statistics of all motifs against their rankings in the identification of TFs specific to the kidney carcinoma class. (b) Mapping the TF activity scores associated with the HNF1A.1 motif to the t-SNE plot. (c) Mapping the expression levels of the *HNF1A* gene to the t-SNE plot. (d) Box plots showing the expression of *HNF1A* in a larger TCGA cohort of patients. TPM, transcripts per million. **Figure S13.** *TP73* is identified as a top-ranked TF for the SC class. (a) Mapping the TF activity scores associated with the TP73 motif to the t-SNE plot. (b) Mapping the expression levels of the *TP73* gene to the t-SNE plot. (c) Box plots showing the expression of *TP73* in the larger TCGA cohort. **Figure S14.** *RFX4* ranks first among the brain cancer class-specific TFs. (a) Plotting the *t*-statistics of all motifs against their rankings in

the identification of TFs specific to the brain cancer class. (b) Mapping the TF activity scores associated with the RFX4 motif to the t-SNE plot. (c) Mapping the expression levels of the *RFX4* gene to the t-SNE plot. (d) Box plots showing the expression of *RFX4* in the larger TCGA cohort as well as in 3,006 RNA-seq samples of normal individuals provided by the GTEx (Genotype-Tissue Expression) project (https://gtexportal.org/home/). We involved the GTEx data because RNA-seq samples for matched normal tissues of GBM and LGG were missing in the TCGA program. **Figure S15.** Examples of stage-specific regulators identified from the mouse ATAC-seq data set. (a, b) Bar plots showing the TF activity scores and expression levels of (a) *Nr5a2* and (b) *Gbx2*, which are top-ranked TFs associated with the early and late stages, respectively. TF activity scores have been averaged across biological replicates for each individual cell stage. **Figure S16.** Evaluating the correlations of HVRs identified by different methods with tumor progression stage. (a, b) Results shown here are based on the LUAD H3K27ac ChIP-seq data set. For each method, the same number of top-ranked proximal/distal HVRs as identified by HyperChIP are selected. The red dotted line in each plot indicates the median correlation of the HVRs identified by HyperChIP. **Figure S17.** Applying different methods to the pan-cancer ATAC-seq data set. (ag) Two-dimensional t-SNE plots generated by different methods. For each method, the same number of top-ranked proximal/distal HVRs as identified by HyperChIP were used for the downstream PCA and t-SNE analysis (see Methods in the main text). (h) Bar plot showing the ARI values achieved by different methods in classifying all samples. For each method, we performed a hierarchical clustering of all samples based on the same principal components as used in the t-SNE analysis. The samples were then classified into 23 sub-groups based on the resulting hierarchical tree, and the corresponding ARI assessed the agreement of this classification with the cancer type labels. The red dotted line indicates the ARI value achieved by HyperChIP. **Figure S18.** Evaluating the ability of different methods to distinguish between the KIRC and KIRP cancer types. (a-g) Zooming in on the t-SNE plots to more clearly present the distributions of KIRC and KIRP samples. (h) Bar plot showing the ARI values achieved by different methods in classifying KIRC and KIRP samples. For each method, we performed a hierarchical clustering of KIRC and KIRP samples and classified them into two sub-groups based on the resulting hierarchical tree. **Figure S19.** Evaluating the ability of different methods to distinguish between the GBM and LGG cancer types. (a-g) Zooming in on the t-SNE plots to more clearly present the distributions of GBM and LGG samples. (h) Bar plot showing the ARI values achieved by different methods in classifying GBM and LGG samples. For each method, we performed a hierarchical clustering of GBM and LGG samples and classified them into two sub-groups based on the resulting hierarchical tree. **Figure S20.** Evaluating the ability of different methods to distinguish between the COAD and STAD cancer types. (a-g) Zooming in on the t-SNE plots to more clearly present the distributions of COAD and STAD samples. (h) Bar plot showing the ARI values achieved by different methods in classifying COAD and STAD samples. For each method, we performed a hierarchical clustering of COAD and STAD samples and classified them into two sub-groups based on the resulting hierarchical tree.

**Additional file 2: Note S1.** Statistical simulation. **Note S2.** Applying HyperChIP without separating proximal and distal regions. **S2.1.** Evaluating separately the rankings of proximal and distal regions. **S2.2.** Evaluating the overall rankings of all peak regions. **Note S3.** Quality control. **Note S4.** Defining proximal and distal regions.

**Additional file 3: Table S1.** Top 20 GO terms enriched from the genes linked with the significant proximal LVRs identified for the H3K27ac ChIP-seq data set. **Table S2.** Top 20 GO terms enriched from the genes linked with the significant proximal HVRs identified for the H3K27ac ChIP-seq data set. **Table S3.** Cancer types involved in the TCGA ATAC-seq data set. Note that in TCGA studies the abbreviation CESC refers to both cervical squamous cell carcinoma and endocervical adenocarcinoma, but all the 4 CESC patients in this data set belong in the former. Note also that the 18 ESCA patients in this data set consist of 12 ESSC (esophageal squamous cell carcinoma) and 6 ESAD (esopha-geal adenocarcinoma) cases, which belong in the SC and DIAD classes, respectively. SC, squamous cell carcinoma; DIAD, digestive adenocarcinoma.

**Additional file 4: Table S4.** Class-specific regulators identified from the TCGA pan-cancer ATAC-seq data set.

**Additional file 5: Table S5.** Stage-specific regulators identified from the mouse ATAC-seq data set.

**Additional file 6: Table S6.** Significant HVRs and LVRs (BH-adjusted *p*-value<0.1) identified by HyperChIP for each of the data sets in Table 1.

**Additional file 7.** Review history.

## Acknowledgements
We sincerely thank the reviewers for their comments and suggestions, which have greatly helped us to further improve the manuscript.

## Peer review information

## Review history
The review history is available as Additional file 7.

## Authors' contributions
S.T. and Z.S. conceived the study. H.C. and S.T. developed the algorithms and analyzed the data. S.T. and Z.S. jointly supervised the study. H.C., S.T., and Z.S. wrote the manuscript with contributions from all the other authors. The authors read and approved the final manuscript.

## Funding

This work was supported by the National Basic Research Program of China (2018YFA0107602 and 2018YFA0800203), the National Natural Science Foundation of China (31871280 and 31701140), and the Strategic Priority Research Program of Chinese Academy of Sciences (XDB38040100).

## Availability of data and materials

The H3K27ac ChIP-seq, ATAC-seq, and Pol II ChIP-seq data sets in Table 1 were obtained from Yuan et al. [6], Wang et al. [18], and Suzuki et al. [32], respectively. Additional file 6: Table S6 gives the significant HVRs and LVRs identified by Hyper-ChIP for these data sets. The pan-cancer ATAC-seq data set was obtained from Corces et al. [17]. The CTCF ChIP-seq data set of human LCLs was obtained from Kasowski et al. [28]. The ATAC-seq data set of mouse preimplantation embryos and mESCs was obtained from Wu et al. [56]. Binding motifs of both human and mouse TFs were obtained from the JASPAR database [45]. The whole HyperChIP model as well as the MA normalization procedure based on a pseudo-reference profile has been implemented and incorporated into MAnorm2 v1.1.0, which was used throughout the whole study. This package has been uploaded to the Zenodo repository (https://doi.org/10.5281/zenodo.5717729) [79]. The latest version of MAnorm2 is always available at GitHub under the GPL-3 license (https://github.com/tushiqi/MAnorm2) [80].

## Declarations

### Ethics approval and consent to participate
Not applicable.

### Consent for publication
Not applicable.

### Competing interests
The authors declare that they have no competing interests.

### Author details
[1]CAS Key Laboratory of Computational Biology, Shanghai Institute of Nutrition and Health, Chinese Academy of Sciences, Shanghai 200031, China. [2]University of Chinese Academy of Sciences, Beijing 100049, China. [3]Department of Thoracic Surgery and State Key Laboratory of Genetic Engineering, Fudan University Shanghai Cancer Center, Shanghai 200032, China. [4]State Key Laboratory of Genetic Engineering, Collaborative Innovation Center of Genetics and Development, Department of Biochemistry, Institute of Plant Biology, School of Life Sciences, Fudan University, Shanghai 200438, China.

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

## 