## [**Additional file 7.** Review history. · Genome Biology]

Review History

First round of review

Reviewer 1

Were you able to assess all statistics in the manuscript, including the appropriateness of statistical tests used? Yes. I assessed the appropriateness of the statistical methods.

Were you able to directly test the methods? No.

Comments to author:

In this manuscript, the authors propose a method HyperChIP for identifying genomic regions with hypervariable signals from ChIP-seq or ATAC-seq samples. Traditionally, most ChIP-seq and ATAC-seq analyses focus on identifying peaks or analyzing differential mean signals between conditions. As ChIP-seq and ATAC-seq studies with a relatively large number of samples become more prevalent, analyzing variability across samples without condition labels become increasingly more important. The HyperChIP method can provide a solution to this problem. The method is based on a statistical model that characterizes hypervariability after accounting for the mean-variance relationship. The authors also develop a method that uses low-intensity regions to estimate a null distribution for determining the statistical significance of the hypervariable signals. The authors have compared HyperChIP with several commonly used variance measures and applied it to analyze a large pan-cancer ATAC-seq dataset to demonstrate its performance and practical utility. Overall, the manuscript is well written. The method and results are logically and clearly presented. The HyperChIP method can add a useful new tool to the ChIP-seq and ATAC-seq data analysis toolbox. I have a few comments below which I hope can help the authors to improve their manuscript.

Major:

1. To infer the null distribution, the unobserved μ_i in formula (3) is replaced by its estimate $\hat{\mu}_i$. Using the estimated μ_i will introduce additional uncertainty so that the ratio between \hat{t}_i and $f(\hat{\mu}_i)$ in theory is no longer an F-distribution. The authors may want to evaluate and/or discuss how this will influence their statistical inference results. Will the F-distribution underestimate the uncertainty/variability and result in optimistic p-values and FDR?
2. The comparisons between HyperChIP and the other methods in Figure 1 and Figure 2 are nice and informative. For the tumor progression analysis in Figure 5F and 5G and for the pan-cancer analysis in Figure 6, it would be interesting to compare HyperChIP with the other methods too (i.e. those methods shown in Figures 1 and 2) to see whether using the same number of HVRs obtained from the other methods can achieve similar results/performance.
3. The authors examined the relationship between single nucleotide variants and HVRs. In tumor samples, copy number variations (CNVs) can be prevalent. CNVs often involve more than one nucleotide. I wonder whether it is feasible here to also analyze the relationship between the hypervariable signals and CNVs. If this analysis is feasible, it would be useful to know to what

extent the hypervariable signals depend on CNV? If it is not feasible, it might be helpful to discuss it. More generally, it will be useful to know whether analyzing hypervariable signals is only useful in cancer studies (where changes in the genomic landscape including genomic DNA are substantial) or it is also useful for studying normal samples.

4. In the discussion, the authors illustrated that proximal and distal regions have different variability (i.e. different γ estimates). It will make the argument much stronger if the authors can show the consequences of not separating proximal and distal regions in the analyses, for example, for the analyses in Figures 1,2,5,6.

Reviewer 2

Were you able to assess all statistics in the manuscript, including the appropriateness of statistical tests used? No.

Were you able to directly test the methods? No.

Comments to author:

In this manuscript, the authors present HyperChIP, a new statistical tool that can identify genomic regions with hypervariable ChIP-seq and ATAC-seq signals across samples. HyperChIP uses scaled variances that account for the mean-variance dependence to rank genomic regions. An advantage of this method over existing methods is that it diminishes the influence of true hypervariable regions (HVRs) on model fitting, which increases the statistical power of the tool for detecting HVRs. The authors applied this method to analyze ChIP-seq and ATAC-seq data across tumors from patient samples in three large datasets to identify hypervariable regions (HRVs) in these samples. Further examination of the HVRs highlighted the need to analyze proximal and distal regions separately to avoid suppression of the statistical power for identifying proximal HRVs, which have lower variability than distal regions. The authors then used HyperChIP to identify HRVs among a pan-cancer ATAC-seq dataset. Together with a motif analysis of the HRVs, they defined classes and super classes of cancer types among the ATAC-seq profiles.

The manuscript could be improved by addressing the following points.

1. In this manuscript, the authors emphasize the need for a statistical method specifically to identify HRVs in human datasets, and indeed demonstrate the utility of HyperChIP to accomplish this task. There is no mention or discussion, however, of the broader need and utility of this tool outside of non-cancer human datasets or in other organisms. For example, it seems that HyperChIP could be used to identify HVRs among large ChIP-seq or ATAC-seq datasets in mouse datasets from heterogeneous cell populations. The authors should discuss possible broader applications of HyperChIP, which may increase utilization of this tool by other research groups.

2. The quality of ChIP-seq datasets is inherently variable, and prior publication of a prior dataset(s) does not guarantee its quality. Although the authors did employ normalization methods

to attempt to account for variations in absolute signal strength as well as signal-to-noise ratios among datasets, low quality datasets may be problematic and confound results. What quality metrics (e.g. FRiP) or standards were used to determine inclusion or exclusion of both ChIP-seq and ATAC-seq datasets used in the analyses?

3. The ChIP-seq datasets selected for evaluation of HyperChIP include histone mark H3K27ac and Pol II, neither of which involve sequence-specific binding of a TF. How does HyperChIP perform on a sequence-specific TF ChIP-seq dataset (e.g. CTCF)? Given the potential utility of HyperChIP in the larger genomic community to define HVRs (or peaks) that have biological significance, evaluation of such dataset would be valuable.

4. The authors define a proximal region as a region with a distance of less than 5kb to a transcription start site (TSS), whereas other regions were deemed to be distal regions. A window \pm 5kb to the TSS seems like an excessively large window, especially since regions up to +5kb of the TSS are not commonly considered to be "proximal". The authors should justify why they chose this particularly large window to define a proximal region.

5. For the mean-variance trends of associated with different datasets, the following graphs are shown:

H3K27ac proximal regions (observed) - Fig 1A

H3K27ac proximal regions (scaled) - Fig 1B

ATAC-seq proximal regions (observed) - Fig 1A

H3K27ac distal regions (observed) - Supp Fig 1

ATAC-seq distal regions (observed) - Supp Fig 1

Pol II proximal regions (observed) - Supp Fig 1

Pol II distal regions (observed) - Supp Fig 1

The authors should include the rest of the graphs in the Supplement, showing the observed and scaled variances for both proximal and distal regions for all three datasets.

6. In Supplementary Figure 4 - For the Pol II proximal regions shown in A and B, the authors should comment on the (potential) discrepancy in the proportion of high intensity regions ranked in the top 2000 HVRs and LVRs. More specifically, why do they see a large rise in the second to last column (left to right) of the higher mean intensity for the HVRs (Supp Fig 4A) and a corresponding dip in the second to last column (left to right) of the higher mean intensity for the LVRs (Supp Fig 4B). This trend seems unusual compared to the other datasets.

7. In Methods, under "Input matrices for normalization", the authors should replace "genders" with "biological sex".

Reviewer #1: In this manuscript, the authors propose a method HyperChIP for identifying genomic regions with hypervariable signals from ChIP-seq or ATAC-seq samples. Traditionally, most ChIP-seq and ATAC-seq analyses focus on identifying peaks or analyzing differential mean signals between conditions. As ChIP-seq and ATAC-seq studies with a relatively large number of samples become more prevalent, analyzing variability across samples without condition labels become increasingly more important. The HyperChIP method can provide a solution to this problem. The method is based on a statistical model that characterizes hypervariability after accounting for the mean-variance relationship. The authors also develop a method that uses low-intensity regions to estimate a null distribution for determining the statistical significance of the hypervariable signals. The authors have compared HyperChIP with several commonly used variance measures and applied it to analyze a large pan-cancer ATAC-seq dataset to demonstrate its performance and practical utility. Overall, the manuscript is well written. The method and results are logically and clearly presented. The HyperChIP method can add a useful new tool to the ChIP-seq and ATAC-seq data analysis toolbox. I have a few comments below which I hope can help the authors to improve their manuscript.

Major:

1. To infer the null distribution, the unobserved μ_i in formula (3) is replaced by its estimate $\hat{\mu}_i$. Using the estimated $\hat{\mu}_i$ will introduce additional uncertainty so that the ratio between \hat{t}_i and $f(\hat{\mu}_i)$ in theory is no longer an F-distribution. The authors may want to evaluate and/or discuss how this will influence their statistical inference results. Will the F-distribution underestimate the uncertainty/variability and result in optimistic p-values and FDR?

Reply: sincerely thanks for this comment. This is a very good point regarding the derivation of the null distribution of scaled variances. We have indeed made an approximation in formula (3) to make the calculation of scaled variances possible, which may increase the uncertainty of the test statistics and may, thus, lead to over-confident p-values. We expect, however, that the overall p-value distribution across genomic regions as well as the specificity of HyperChIP is resistant to the approximation, since the whole model fitting process is based on the same approximation as well. More specifically, it is $\hat{\mu}_i$ rather than μ_i that is used in the MVC fitting and parameter estimation procedures. Accordingly, the aim of these procedures is to make the resulting model fit the observed $\hat{t}_i/f(\hat{\mu}_i)$ rather than the unobserved $\hat{t}_i/f(\mu_i)$. In fact, $\hat{\mu}_i$ is used as an ordinary covariate for regressing variances throughout the procedures, though it is not strictly non-stochastic.

To verify our speculation, we performed a series of statistical simulation in which formulas (1) and (2) were used as the data generation process, and we found that

the overall p -value distribution was very uniform on [0, 1] across various scenarios. Please refer to Note S1 in Additional file 2 for the simulation results as well as a detailed discussion of the topic.

2. The comparisons between HyperChIP and the other methods in Figure 1 and Figure 2 are nice and informative. For the tumor progression analysis in Figure 5F and 5G and for the pan-cancer analysis in Figure 6, it would be interesting to compare HyperChIP with the other methods too (i.e. those methods shown in Figures 1 and 2) to see whether using the same number of HVRs obtained from the other methods can achieve similar results/performance.

Reply: sincerely thanks for this suggestion. We have accordingly performed new analyses and have added a paragraph in Discussion (the 2nd one; page 15) to discuss the influence of the specific hypervariable analysis method on various downstream analyses. In detail, we first applied all the other methods to the LUAD H3K27ac ChIP-seq data set (the same number of top-ranked proximal/distal HVRs as identified by HyperChIP were selected for each method). For these methods, the correlations of selected HVRs with tumor progression stage were either roughly as strong as observed from HyperChIP or weaker (Additional file 1: Fig. S16). Then, we applied these methods to the pan-cancer ATAC-seq data set and repeated the t-SNE analysis as presented in Figure 6a. On the one hand, the two-dimensional t-SNE plots generated by different methods exhibited similar structures (Additional file 1: Fig. S17). On the other hand, we noticed that HyperChIP performed better in revealing fine structures among the samples. In particular, it showed an ability to more accurately distinguish between the cancer types belonging in the same super class. Examples included the KIRC and KIRP types of the kidney carcinoma class, the GBM and LGG types of the brain cancer class, and the COAD and STAD types of the DIAD class (Additional file 1: Figs. S18-S20). Together, these results further suggested the stable performance of HyperChIP.

3. The authors examined the relationship between single nucleotide variants and HVRs. In tumor samples, copy number variations (CNVs) can be prevalent. CNVs often involve more than one nucleotide. I wonder whether it is feasible here to also analyze the relationship between the hypervariable signals and CNVs. If this analysis feasible, it would be useful to know to what extent the hypervariable signals depend on CNV? If it is not feasible, it might be helpful to discuss it. More generally, it will be useful to know whether analyzing hypervariable signals is only useful in cancer studies (where changes in the genomic landscape including genomic DNA are substantial) or it is also

useful for studying normal samples.

Reply: sincerely thanks for this valuable comment. It is really interesting to explore whether hypervariable ChIP/ATAC-seq analysis is useful in biological contexts where the associated genomic variation is not as substantial as in cancer studies.

A CNV analysis was performed based on the NSCLC ATAC-seq data set as well. We first considered an ATAC-seq peak region as associated with somatic CNV only if it overlapped CNV segments in more than 5 patients, since the CNV segments identified from all the NSCLC patients together occupied almost the whole genome (>95%). Even under this cutoff, a considerable proportion of the identified HVRs were found to be associated with somatic CNV (31.8% and 28.7% for the proximal and distal HVRs, respectively), demonstrating a clear association between the hypervariable ATAC-seq signals and CNV. Moreover, both the two proportions were significantly higher than observed from randomly selected peak regions (Additional file 1: Fig. S8).

We next applied HyperChIP to two non-cancer data sets. The first data set can serve as an example for studying the variation in TF binding intensities across normal humans. This data set consisted of CTCF ChIP-seq samples of 17 lymphoblastoid cell lines (LCLs) derived from different human individuals, including 6 Caucasian individuals, 7 Yoruban individuals, and 4 individuals from the San population. Compared with the cancer data sets, the number of significant HVRs identified from this data set was much smaller (364 proximal HVRs and 498 distal ones with BH-adjusted p -values less than 0.1; see also Tables 2 and 3). We then performed principal component analysis (PCA) of the samples with all CTCF peak regions or only the HVRs as features. Interestingly, while the LCLs from different populations were mixed together in the former case, these LCLs were well clustered by their populations of origin in the latter case (Fig. 7a, b). This finding suggested that the hypervariable CTCF binding signals captured by HyperChIP across the LCLs were useful for dissecting the similarity structure among them.

The second data set comprised ATAC-seq samples of mouse preimplantation embryos at different stages (the 2-cell, 4-cell, 8-cell embryos and the inner cell masses (ICMs) of the blastocysts) and mouse embryonic stem cells (mESCs; derived from ICMs). This data set can serve as a good example to illustrate the utility of hypervariable analysis for samples with temporal labels. Applying HyperChIP, we identified 303 proximal HVRs and 383 distal ones (BH-adjusted p -value<0.1). PCA with these HVRs as features revealed that a large proportion (71.6%) of the ATAC-seq signal variability at these regions was accounted for by the first principal component, which showed a very strong association with the development timeline (Fig. 7c). We then accordingly classified the samples into early-stage and late-stage ones, and we repeated the motif analysis applied to the pan-cancer ATAC-seq data set to identify stage-specific regulators (Fig. 7c, d). The identified regulators were largely consistent

with previous reports as well as their gene expression profiles (Fig. 7d-f; Additional file 1: Fig. S15). Please refer to the section of “Applying HyperChIP to non-cancer data sets” (page 13) for details.

Together, these findings indicated the usefulness of hypervariable analysis for ChIP/ATAC-seq samples from normal tissues/cells.

4. In the discussion, the authors illustrated that proximal and distal regions have different variability (i.e. different γ estimates). It will make the argument much stronger if the authors can show the consequences of not separating proximal and distal regions in the analyses, for example, for the analyses in Figures 1,2,5,6.

Reply: thanks for this nice suggestion. We have accordingly applied HyperChIP to the identification of HVRs without separating proximal and distal regions. For simplicity, we refer to this analysis strategy as the combined method, and refer to the original strategy as the separated method.

Since proximal and distal regions are typically associated with distinct mean-variance trends (Additional file 1: Fig. S1), we expect that the MVC fitting procedure of the combined method has to compromise between the two classes of regions, which will influence not only the overall rankings of all peak regions but also the rankings within each class. To explore this speculation, we separately evaluated the rankings of proximal and distal regions when comparing the performance of the two methods. Applying the combined method to the data sets in Table 1, we found that the rankings of proximal/distal regions became worse with respect to the consistency with HVGs and the resulting classifications of the NSCLC ATAC-seq samples (Additional file 2: Note S2.1).

We also examined the overall rankings of all peak regions when comparing the two methods. In the classification analysis of the NSCLC ATAC-seq samples, we ranked together proximal and distal regions based on either raw p -values or BH-adjusted ones when applying the separated method. It was found that the separated method outperformed the combined method in both cases (Additional file 2: Note S2.2). For the t-SNE analysis of the pan-cancer ATAC-seq data set, we have shown the results from the application of the separated method (Fig. 6a), in which a BH-adjusted p -value cutoff of 0.1 was applied to the selection of both proximal and distal HVRs. In effect, this was equivalent to combining the rankings of proximal and distal regions based on BH-adjusted p -values and selecting a certain number of top-ranked HVRs. Accordingly, we applied the combined method to the same analysis with selecting the same total number of top-ranked HVRs. Again, the two-dimensional t-SNE plots generated by the two methods were similar to each other, but the separated method performed better in distinguishing between the cancer types belonging to the same super class. Please

refer to Note S2.2 in Additional file 2 for details.

We also applied the combined method to the analysis presented in Figure 5f, g for checking the correlations of identified HVRs with tumor progression stage. It was found that the performance was very similar to that of the separated method:

For the left two plots, we selected the same number of top-ranked proximal/distal HVRs (as identified by the separated method) when applying the combined method; for the rightmost one, the same total number of top-ranked HVRs were selected.

Overall, we believe these findings further indicate the necessity of separating proximal and distal regions in hypervariable ChIP/ATAC-seq analysis.

Reviewer #2: To authors:

In this manuscript, the authors present HyperChIP, a new statistical tool that can identify genomic regions with hypervariable ChIP-seq and ATAC-seq signals across samples. HyperChIP uses scaled variances that account for the mean-variance dependence to rank genomic regions. An advantage of this method over existing methods is that it diminishes the influence of true hypervariable regions (HVRs) on model fitting, which increases the statistical power of the tool for detecting HVRs. The authors applied this method to analyze ChIP-seq and ATAC-seq data across tumors from patient samples in three large datasets to identify hypervariable regions (HRVs) in these samples. Further examination of the HVRs highlighted the need to analyze proximal and distal regions separately to avoid suppression of the statistical power for identifying proximal HRVs, which have lower variability than distal regions. The authors then used HyperChIP to identify HRVs among a pan-cancer ATAC-seq dataset. Together with a motif analysis of the HRVs, they defined classes and super classes of cancer types among the ATAC-seq profiles.

The manuscript could be improved by addressing the following points.

1. In this manuscript, the authors emphasize the need for a statistical method specifically to identify HRVs in human datasets, and indeed demonstrate the utility of HyperChIP to accomplish this task. There is no mention or discussion, however, of the broader need and utility of this tool outside of non-cancer human datasets or in other organisms. For example, it seems that HyperChIP could be used to identify HVRs among large ChIP-seq or ATAC-seq datasets in mouse datasets from heterogeneous cell populations. The authors should discuss possible broader applications of HyperChIP, which may increase utilization of this tool by other research groups.

Reply: sincerely thanks for this valuable comment. It is really interesting and important to explore the practical utility of HyperChIP in non-cancer and/or non-human contexts.

Following your suggestion, we applied HyperChIP to two non-cancer data sets (one of them was generated from mouse cells). The first data set can serve as an example for studying the variation in TF binding intensities across normal humans. This data set consisted of CTCF ChIP-seq samples of 17 lymphoblastoid cell lines (LCLs) derived from different human individuals, including 6 Caucasian individuals, 7 Yoruban individuals, and 4 individuals from the San population. Applying HyperChIP, we identified 364 proximal HVRs and 498 distal ones (BH-adjusted p -value <0.1). We then performed principal component analysis (PCA) of the samples with all CTCF peak regions or only the HVRs as features. Interestingly, while the LCLs from different populations were mixed together in the former case, these LCLs were well clustered by their populations of origin in the latter case (Fig. 7a, b). This finding suggested that

the hypervariable CTCF binding signals captured by HyperChIP across the LCLs were useful for dissecting the similarity structure among them.

The second data set comprised ATAC-seq samples of mouse preimplantation embryos at different stages (the 2-cell, 4-cell, 8-cell embryos and the inner cell masses (ICMs) of the blastocysts) and mouse embryonic stem cells (mESCs; derived from ICMs). This data set can serve as a good example to illustrate the utility of hypervariable analysis for samples with temporal labels. Applying HyperChIP, we identified 303 proximal HVRs and 383 distal ones (BH-adjusted p -value <0.1). PCA with these HVRs as features revealed that a large proportion (71.6%) of the ATAC-seq signal variability at these regions was accounted for by the first principal component, which showed a very strong association with the development timeline (Fig. 7c). We then accordingly classified the samples into early-stage and late-stage ones, and the same motif analysis as applied to the pan-cancer ATAC-seq data set was repeated to identify stage-specific regulators (Fig. 7c, d). The identified regulators were largely consistent with previous reports as well as their gene expression profiles (Fig. 7d-f; Additional file 1: Fig. S15). Please refer to the section of “Applying HyperChIP to non-cancer data sets” (page 13) for details.

Together, these analyses demonstrated the utility of HyperChIP in analyzing non-cancer and/or non-human samples.

2. The quality of ChIP-seq datasets is inherently variable, and prior publication of a prior dataset(s) does not guarantee its quality. Although the authors did employ normalization methods to attempt to account for variations in absolute signal strength as well as signal-to-noise ratios among datasets, low quality datasets may be problematic and confound results. What quality metrics (e.g. FRiP) or standards were used to determine inclusion or exclusion of both ChIP-seq and ATAC-seq datasets used in the analyses?

Reply: sincerely thanks for this comment. We have indeed examined the FRiPs of samples for each data set (Additional file 2: Note S3). On the one hand, the distribution of FRiPs varied considerably across different data sets, owing to various biological and technical reasons. For example, the mouse ATAC-seq data set was associated with much lower FRiPs compared to the other data sets, due to the low input materials obtained from preimplantation embryos [1]. On the other hand, none of the data sets was associated with outlier samples that had extremely low FRiP (or peak number) compared to the other samples in the same data set. Therefore, we basically retained every sample but only filtered out the NSCLC ATAC-seq samples with less than 40k peaks, which was based on our previous experience in analyzing ATAC-seq data sets from cancer studies.

For the sake of rigor, we also performed the related benchmarking analyses without filtering out any NSCLC ATAC-seq samples. Overall, the performance of all the involved methods was worse than before, but HyperChIP still performed relatively better than the other methods. Please refer to Note S3 in Additional file 2 for details.

3. The ChIP-seq datasets selected for evaluation of HyperChIP include histone mark H3K27ac and Pol II, neither of which involve sequence-specific binding of a TF. How does HyperChIP perform on a sequence-specific TF ChIP-seq dataset (e.g. CTCF)? Given the potential utility of HyperChIP in the larger genomic community to define HVRs (or peaks) that have biological significance, evaluation of such dataset would be valuable.

Reply: sincerely thanks for this valuable comment. We have accordingly incorporated a new data set consisting of CTCF ChIP-seq samples of 17 LCLs derived from different human individuals. Please refer to the first point as well as the section of “Applying HyperChIP to non-cancer data sets” (page 13) for details.

4. The authors define a proximal region as a region with a distance of less than 5kb to a transcription start site (TSS), whereas other regions were deemed to be distal regions. A window \pm 5kb to the TSS seems like an excessively large window, especially since regions up to +5kb of the TSS are not commonly considered to be “proximal”. The authors should justify why they chose this particularly large window to define a proximal region.

Reply: sincerely thanks for this comment. We propose the separation of proximal and distal regions in hypervariable ChIP/ATAC-seq analysis because the global ChIP/ATAC-seq signal variability in distal regions is typically higher than that in proximal regions (Fig. 8). A primary reason accounting for this difference is that the activity of distal regulatory elements is much more variable across cellular contexts and human individuals than is gene expression [2-4], while the activity of proximal ones is tightly connected with the expression of nearby genes. The use of a 5kb window to define proximal regions is exactly for capturing such regulatory elements whose activity is strongly correlated with the expression of nearby genes.

Specifically, we examined, for each data set in Table 1, the Pearson correlation coefficient (PCC) between the ChIP/ATAC-seq signal intensities in each peak region and the expression of the nearest gene (\log_2 -CPM values calculated from RNA-seq samples). We then performed a regression of the PCC values (by applying LOWESS: locally-weighted polynomial regression [5]) against the distances between peak regions and genes (Additional file 2: Note S4). For each data set, the fitted PCC

achieved its highest value at the distance of 0, which corresponded to the peak regions occupying a TSS, and it gradually decreased as the distance became larger. We noticed that the peak-gene association was still strong up to 5kb away from TSS. In particular, the fitted PCC at 5kb was within 0.1 of the highest value for each data set. Moreover, for both the H3K27ac ChIP-seq and ATAC-seq data sets, the fitted PCC curves had break points near 5kb (the curve associated with the Pol II ChIP-seq data set had a break point less than 2kb). We therefore have chosen 5kb as the distance cutoff for separating proximal and distal regions.

For the sake of rigor, we also tried using 2kb as the distance cutoff and repeated the benchmarking analyses as presented in Figure 1c, d and Figure 2. Overall, the performance of HyperChIP was better and more stable than all the other methods.

5. For the mean-variance trends of associated with different datasets, the following graphs are shown:

H3K27ac proximal regions (observed) - Fig 1A

H3K27ac proximal regions (scaled) - Fig 1B

ATAC-seq proximal regions (observed) - Fig 1A

H3K27ac distal regions (observed) - Supp Fig 1

ATAC-seq distal regions (observed) - Supp Fig 1

Pol II proximal regions (observed) - Supp Fig 1

Pol II distal regions (observed) - Supp Fig 1

The authors should include the rest of the graphs in the Supplement, showing the observed and scaled variances for both proximal and distal regions for all three datasets.

Reply: thanks for this nice suggestion. We have accordingly filled the missing graphs. The Figures S1 and S2 in Additional file 1 shows the observed and scaled variances for all cases, respectively.

6. In Supplementary Figure 4 - For the Pol II proximal regions shown in A and B, the authors should comment on the (potential) discrepancy in the proportion of high intensity regions ranked in the top 2000 HVRs and LVRs. More specifically, why do they see a large rise in the second to last column (left to right) of the higher mean intensity for the HVRs (Supp Fig 4A) and a corresponding dip in the second to last column (left to right) of the higher mean intensity for the LVRs (Supp Fig 4B). This trend seems unusual compared to the other datasets.

Reply: sincerely thanks for this comment. For the Pol II ChIP-seq data set, we carefully

examined the distributions of top-ranked proximal HVRs and LVRs in the mean-variance scatter plot (Additional file 1: Fig. S6). It was found that the top-ranked LVRs form two clusters that are somewhat separated from one another in the plot, owing to a gap largely corresponding to the 70th to 90th percentile of mean intensities. As a result, the proportion of the LVRs dips at the corresponding two groups of regions, leading to a bimodal distribution profile as well as a rise in the HVR proportion that is more dramatic compared to the other two data sets. Since this pattern is not very typical of practical ChIP/ATAC-seq data (according to our experience) and it does not essentially conflict with our model fitting strategy, which is to use only low-intensity regions to estimate parameters, we did not explore it in depth but explicitly pointed it out and commented on it in the manuscript (page 8).

7. In Methods, under "Input matrices for normalization", the authors should replace "genders" with "biological sex".

Reply: thanks for this nice reminder. We have accordingly made a modification in the section (page 18).

References

1. Wu J, Huang B, Chen H, Yin Q, Liu Y, Xiang Y, Zhang B, Liu B, Wang Q, Xia W, et al: **The landscape of accessible chromatin in mammalian preimplantation embryos.** *Nature* 2016, **534**:652-657.
2. Ernst J, Kheradpour P, Mikkelson TS, Shores N, Ward LD, Epstein CB, Zhang X, Wang L, Issner R, Coyne M, et al: **Mapping and analysis of chromatin state dynamics in nine human cell types.** *Nature* 2011, **473**:43-49.
3. Kasowski M, Kyriazopoulou-Panagiotopoulou S, Grubert F, Zaugg JB, Kundaje A, Liu Y, Boyle AP, Zhang QC, Zakharia F, Spacek DV, et al: **Extensive variation in chromatin states across humans.** *Science* 2013, **342**:750-752.
4. Heinz S, Romanoski CE, Benner C, Glass CK: **The selection and function of cell type-specific enhancers.** *Nat Rev Mol Cell Biol* 2015, **16**:144-154.
5. Cleveland WS: **Lowess - a Program for Smoothing Scatterplots by Robust Locally Weighted Regression.** *American Statistician* 1981, **35**:54-54.

Second round of review

Reviewer 1

The authors have satisfactorily addressed my previous questions.

Reviewer 2

The authors have sufficiently addressed my concerns. I think HyperChIP will be a valuable tool for analyzing of hypervariable signals across both ChIP and ATAC samples.